# Tissue-specific profiling of age-dependent miRNAomic changes in *Caenorhabditis elegans*

Xueqing Wang[1,2,7], Quanlong Jiang[3,4,7], Hongdao Zhang [1,2,7], Zhidong He[1,2], Yuanyuan Song[1,2], Yifan Chen[1,2], Na Tang[1,2], Yifei Zhou [1,2], Yiping Li[1,2], Adam Antebi[5,6], Ligang Wu [1,2] ✉, Jing-Dong J. Han [4] ✉ & Yidong Shen [1,2] ✉

Ageing exhibits common and distinct features in various tissues, making it critical to decipher the tissue-specific ageing mechanisms. MiRNAs are essential regulators in ageing and are recently highlighted as a class of intercellular messengers. However, little is known about the tissue-specific transcriptomic changes of miRNAs during ageing. *C. elegans* is a well-established model organism in ageing research. Here, we profile the age-dependent miRNAomic changes in five isolated worm tissues. Besides the diverse ageing-regulated miRNA expression across tissues, we discover numerous miRNAs in the tissues without their transcription. We further profile miRNAs in the extracellular vesicles and find that worm miRNAs undergo inter-tissue trafficking via these vesicles in an age-dependent manner. Using these datasets, we uncover the interaction between body wall muscle-derived *mir-1* and DAF-16/FOXO in the intestine, suggesting *mir-1* as a messenger in inter-tissue signalling. Taken together, we systematically investigate worm miRNAs in the somatic tissues and extracellular vesicles during ageing, providing a valuable resource to study tissue-autonomous and nonautonomous functions of miRNAs in ageing.

Ageing manifests with a systematic decline of physiological functions across tissues. After decades of research, it is clear that ageing is under complex and delicate control of intracellular and intercellular signalling[1,2]. Proper manipulations of these signalling effectively extend both the healthspan and lifespan of animals[1–3]. Therefore, it is vital to identify the molecular mechanisms underlying the longevity signals. Furthermore, it is crucial to decipher the ageing signalling in individual tissues since different tissues exhibit common and specific ageing phenotypes with diverse mRNA transcriptomic changes[4–7].

MicroRNAs (miRNAs) are a class of small RNA, which suppress gene expression by binding to the 3′-UTR of mRNAs. Whereas early studies focused on their intracellular functions, growing evidence indicates that secreted miRNAs encapsulated in extracellular vesicles (EVs) also act as messengers in intercellular communication[8,9]. As in many other biological activities, numerous miRNAs have been identified to be critical regulators of ageing across taxa. However, previous studies focused on individual miRNAs or miRNAomic changes at an organismal level[10–13]. Little is known about the tissue-specific changes of miRNAome in ageing.

[1]State Key Laboratory of Cell Biology, Shanghai Institute of Biochemistry and Cell Biology, Center for Excellence in Molecular Cell Science, Chinese Academy of Sciences, 200031 Shanghai, China. [2]University of Chinese Academy of Sciences, 100049 Beijing, China. [3]CAS Key Laboratory of Computational Biology, Shanghai Institute of Nutrition and Health, Shanghai Institutes for Biological Sciences, Chinese Academy of Sciences, 200031 Shanghai, China. [4]Peking-Tsinghua Center for Life Sciences, Academy for Advanced Interdisciplinary Studies, Center for Quantitative Biology (CQB), Peking University, 102213 Beijing, China. [5]Max Planck Institute for Biology of Ageing, D-50931 Cologne, Germany. [6]Cologne Excellence Cluster on Cellular Stress Responses in Aging-Associated Diseases (CECAD), University of Cologne, 50674 Cologne, Germany. [7]These authors contributed equally: Xueqing Wang, Quanlong Jiang, Hongdao Zhang. ✉e-mail: lgwu@sibcb.ac.cn; jackie.han@pku.edu.cn; yidong.shen@sibcb.ac.cn

MiRNAs are also critical intercellular messengers in addition to their intracellular functions[14]. Notably, the levels of miRNAs in human serum exhibit significant changes during ageing[15,16]. Moreover, we recently discovered that intestinal miR-83 induces the age-related decrease of autophagy in the body wall muscle of *Caenorhabditis elegans*[13]. These findings highlight the potential role of miRNAs in controlling ageing across tissues. However, a systematic view of miRNAs in tissue-tissue ageing signalling is yet to be explored.

*C. elegans* is a well-established model organism in ageing research. With a short lifespan of weeks, a simple body architecture, and conserved ageing phenotypes and mechanisms, it has enabled numerous findings in the biology of ageing[17]. Meanwhile, *C. elegans* also pioneers the study of miRNA biology[18,19]. Nevertheless, due to its tiny body size, the previous age-dependent miRNAomic studies in *C. elegans* were from the whole worm[13,20,21], thus blurring the role of tissue-specific and inter-tissue miRNA signalling in ageing.

We recently established a method to profile transcriptomes in isolated worm tissues[4,13]. Here, using this method, we report the miRNAomic changes in five major worm somatic tissues during ageing. We further examined the age-dependent miRNA transcription in these tissues using GFP reporters driven by miRNA promoters. We found many miRNAs in the tissues without transcription, implying a complex miRNA trafficking network across worm tissues. In line, we discovered that worm miRNAs undergo age-dependent inter-tissue trafficking mediated by EVs. Taken together, these datasets showed that ageing controls miRNAs in a tissue-specific manner to modulate gene expression. Moreover, our findings suggest a complex EV-mediated miRNA trafficking network across worm tissues, which could coordinate ageing throughout the body.

## Results

### Ageing changes miRNAomes in *C. elegans* tissues

To examine the age-dependent miRNAomic changes in worm tissues, we collected young worms on day 1 of adulthood (D1) and aged worms post-reproduction on day 8 of adulthood (D8) and isolated cells from five major somatic tissues (i.e., neurons, intestine, body wall muscle, hypodermis, and coelomocyte) by tissue-specific fluorescent markers[4]. RNA from the isolated cells was divided into two aliquots. One was subjected to Smart-Seq2 for the tissue-specific mRNA expression during ageing, as we previously reported[4,22], and the other to miRNA-Seq in this study[13,23] (Fig. 1a). Cell purity and viability were validated to exclude potential contamination from fragments of other tissues or dead cells, as demonstrated in our previous report[4].

By the curations in MirGeneDB[24], our tissue-specific miRNA-Seq identified 144 mature miRNAs from 98 miRNA genes in 64 families (Supplementary Data 1). Our results overlap with three previous miRNAomic studies of worm tissues with more miRNAs identified (Supplementary Fig. 1, Supplementary Data 2)[25–27]. Of the miRNAs reported in miRbase, 113 are considered highly confident[28], and we detected 83 of them. Our dataset thus should cover the majority of known worm miRNAs in the tested tissues, considering that some miRNAs may not be expressed in these tissues[29]. Due to the limited samples for miRNA-Seq, miRNA isoforms were not analysed[30,31].

As anticipated, the miRNAomes exhibited significant changes in ageing tissues. 80 of the 144 identified miRNAs showed significant alterations in at least one examined tissue during ageing and were defined as age-dependent differentially expressed miRNAs (Age-DEMIRs). When focusing on individual tissues, 38.4%, 31.3%, 29.4%, 36.4%, and 30.4% miRNAs underwent age-dependent changes in neurons, intestine, body wall muscle (BWM), hypodermis, and coelomocytes, respectively (Fig. 1b). Many ageing phenotypes and mechanisms in worms are similar to those in mammals[4,17]. Consistently, 52.5% of Age-DEMIRs are conserved in mammals (Supplementary Data 1). From a transcriptomic perspective, neurons, intestine, and hypodermis were clustered first by their tissue types and then by their ages, whereas

BWM and coelomocytes were first clustered by their ages and then by their tissue types (Fig. 1c, d), suggesting different ageing rates of miRNAomes in these tissues. Consistent with the fact that different worm tissues share many biological processes[4], principal component analysis also implies similarities in these tissues (Fig. 1c).

In a previous study, we examined the age-dependent changes in the miRNAome of whole worms[13]. The Age-DEMIRs in the whole worm showed limited overlaps with tissue-specific Age-DEMIRs (Fig. 2a). Consistently, the examined tissues exhibited clear tissue-specificities in their Age-DEMIRs (Fig. 2b, c). For example, no upregulated Age-DEMIRs were shared across the five tissues, and only two miRNAs were found to decrease in all the tissues with ageing (Fig. 2b, c). The tissue-specificity of Age-DEMIRs is not attributed to the selective expression of miRNAs in worm tissues because we observed a strong overlap of miRNA expression across tissues (Supplementary Fig. 2). Thus, ageing alters miRNAome in a tissue-specific manner.

### MiRNA abundance in tissues is inconsistent with the activity of their promoters

Previous research examined miRNA expression in young worm tissues using GFP reporters driven by miRNA promoters (*PmiR::GFP*)[29]. Surprisingly, our tissue-specific miRNA-Seq identified many miRNAs in the tissues without a reported *PmiR::GFP* signal (Supplementary Data 1)[29]. Promoter activity is directly linked to transcription, whereas miRNA levels are also regulated by intercellular transfer, as in the case of *mir-83*[13], in addition to intracellular transcription. Therefore, we speculate that this discrepancy could be from the take-in of secreted miRNAs from other tissues.

To test this hypothesis, we systematically checked miRNA transcription in young and aged worms, using *PmiR::GFP* transgenes in published reports[20,29,32]. As previous studies did not examine these transgenes in aged worms[20,29,32], we checked their expression at day 1 and day 8 of adulthood by ourselves. In our detected 98 miRNA genes, 63 are driven by 56 presumed independent promoters, which were reported to confer GFP expression in any of the five examined tissues[20,29,32]. Among them, 11 miRNA genes with *PmiR::GFP* signal were not detected by our tissue-specific miRNA-Seq. This could be due to their low expression in the examined tissues at D1 or D8. Besides, *PmiR::GFP* may fail to reflect their endogenous transcription because 4 of the 11 miRNA genes were not detected in previous miRNAomic analysis in worm tissues (Supplementary Data 2)[20,25–27]. To avoid any potential ambiguity, these miRNAs were therefore excluded from our analysis. We detected 82 mature miRNAs from the remaining 52 miRNA genes by both miRNA-Seq and *PmiR::GFP*, and subjected them to further analyses.

The transcription of analysed miRNAs exhibited strong tissue-specificities as reported[29] (Fig. 3a). We found a considerable difference in miRNA transcription and abundance, consistent with the situation where inter-tissue miRNA transport controls miRNAomes across tissues. 69 mature miRNAs were detected by miRNA-Seq but not by their *PmiR::GFP* reporters in at least one examined tissue (Fig. 3a), implying that these miRNAs could be taken in from other tissues. These 69 miRNAs were hereafter referred to as Potentially Inter-Tissue Transported miRNAs (PITT-miRs). Moreover, the inconsistency between miRNA-Seq result and *PmiR::GFP* signal in worm tissues changed with ageing (Fig. 3a), suggesting that ageing alters inter-tissue miRNA trafficking. On the other hand, 60.9% PITT-miRs are Age-DEMIRs (Fig. 3b), implying that inter-tissue miRNA signalling also regulates ageing.

### Ageing controls inter-tissue miRNA trafficking

MiRNAs not only function in the cell of their transcription but are also transported from cell to cell as signalling messengers[33]. We investigated miRNA trafficking across the five tested tissues in young and aged worms based on the PITT-miRs. A PITT-miRNA is considered to be

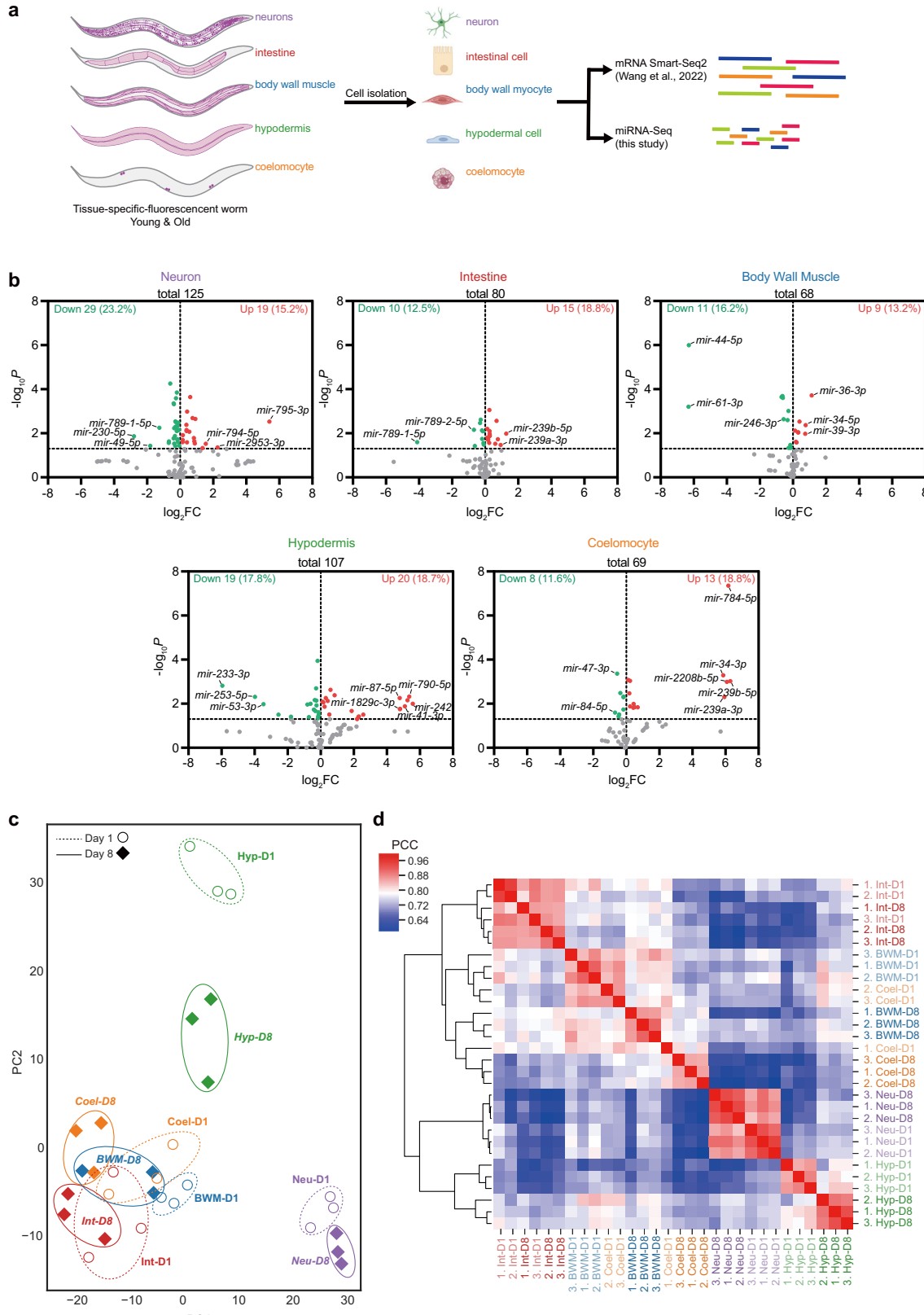

**Fig. 1 | The tissue-specific changes of miRNA transcriptomes during worm ageing. a** A depiction of the tissue-specific analysis of miRNA transcriptomes during worm ageing. Created with BioRender.com. **b** The volcano plots showing the differentially expressed miRNAs in indicated ageing tissues. MiRNAs with a $P$ value smaller than 0.05 were considered significantly changed by ageing. Two-sided $t$-test without adjustments for multiple comparisons. **c**, **d** Principal component analysis (**c**) and hierarchical clustering (**d**) of the miRNAomic datasets of indicated tissues from the worms at day 1 (D1) or day 8 (D8) of adulthood. *Neu* neuron, *Int* intestine, *BWM* body wall muscle, *Hyp* hypodermis, *Coel* coelomocyte.

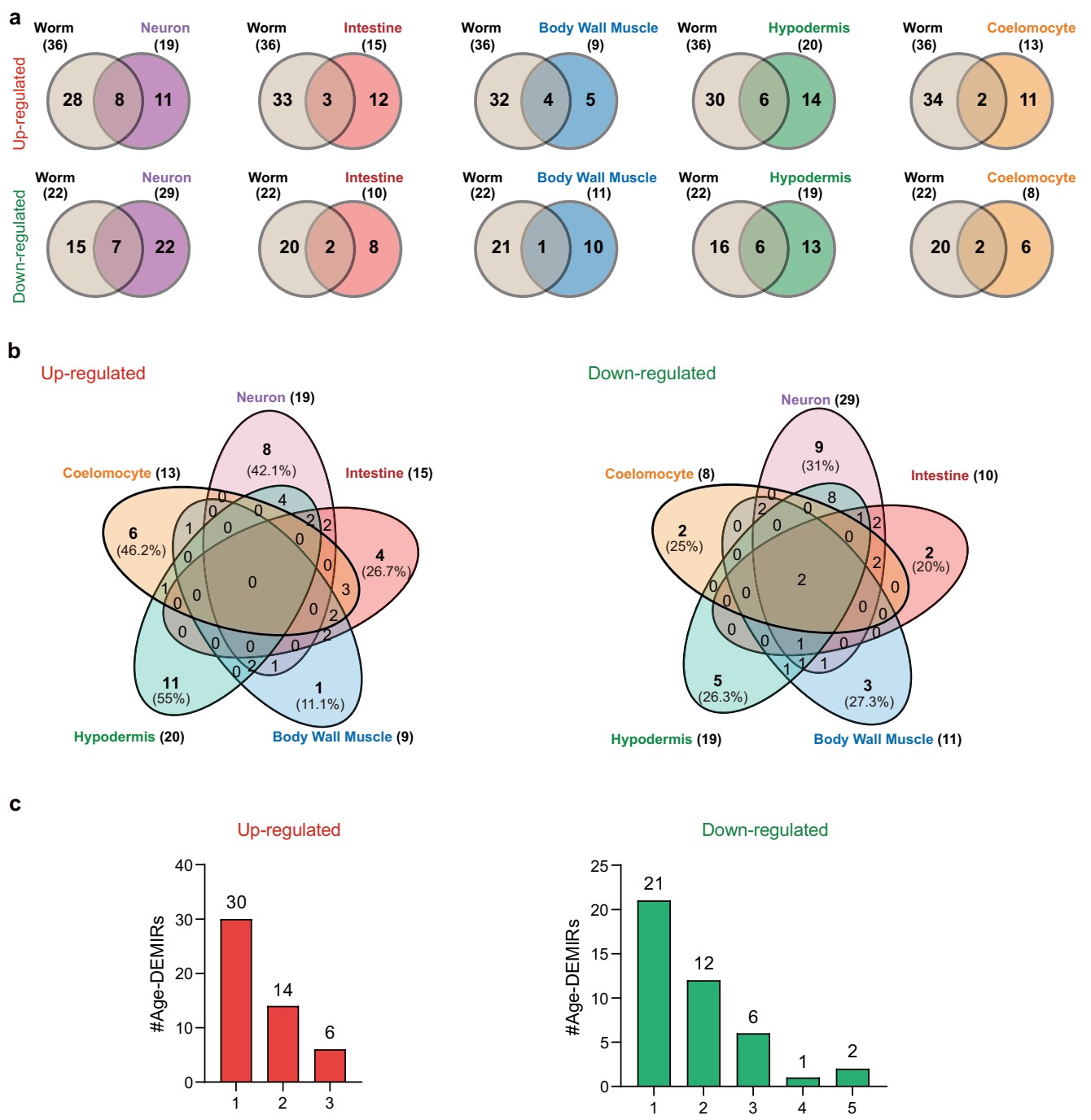

**Fig. 2 | The comparison of ageing-regulated miRNAs across examined tissues. a** The Venn diagrams show the overlap of Age-DEMIRs from the whole worm and the indicated tissues. **b** The Venn diagrams of Age-DEMIRs in the five examined tissues. **c** The shared Age-DEMIRs decreases with the increase of analysed tissues.

transported from tissue A to B if it is identified in A by both miRNA-Seq and *PmiR::GFP*, whereas detected in B by miRNA-Seq but not by *PmiR::GFP* (Fig. 4a). Our datasets then suggested a complex network of tissue-tissue miRNA communication in the worm (Fig. 4b, c). PITT-miRs could be transported between each pair of examined tissues, and one tissue could take in the same PITT-miRs from multiple other tissues. Besides, our analysis implies that PITT-miRs could be transported from other unanalysed tissues into these examined ones. In both young and aged worms, the miRNA flow from coelomocyte to intestine was significantly weaker, whereas the one from neuron to hypodermis was significantly stronger than the others at D8 (Fig. 4d). Besides, neurons and intestine acted more as sources, whereas BWM and

coelomocytes were more like receivers of inter-tissue miRNA signalling (Fig. 4d, e). It is worth noting that inter-tissue miRNA transport is not unidirectional. For example, even in neurons which is a significant source of secreted miRNAs, 41.9% and 38.2% of miRNAs could be from other tissue (e.g., hypodermis) as PITT-miRs in young and aged worms, respectively (Fig. 4b, c).

Interestingly, ageing altered this inter-tissue miRNA trafficking network (Fig. 4b–e). In the five examined tissues, hypodermis underwent the most significant change in miRNA trafficking during ageing. It sent out fewer miRNAs while receiving more at D8 (Fig. 4e). Although the transported miRNAs among young and aged worm tissues largely overlap, their composition is still different (Supplementary Fig. 3).

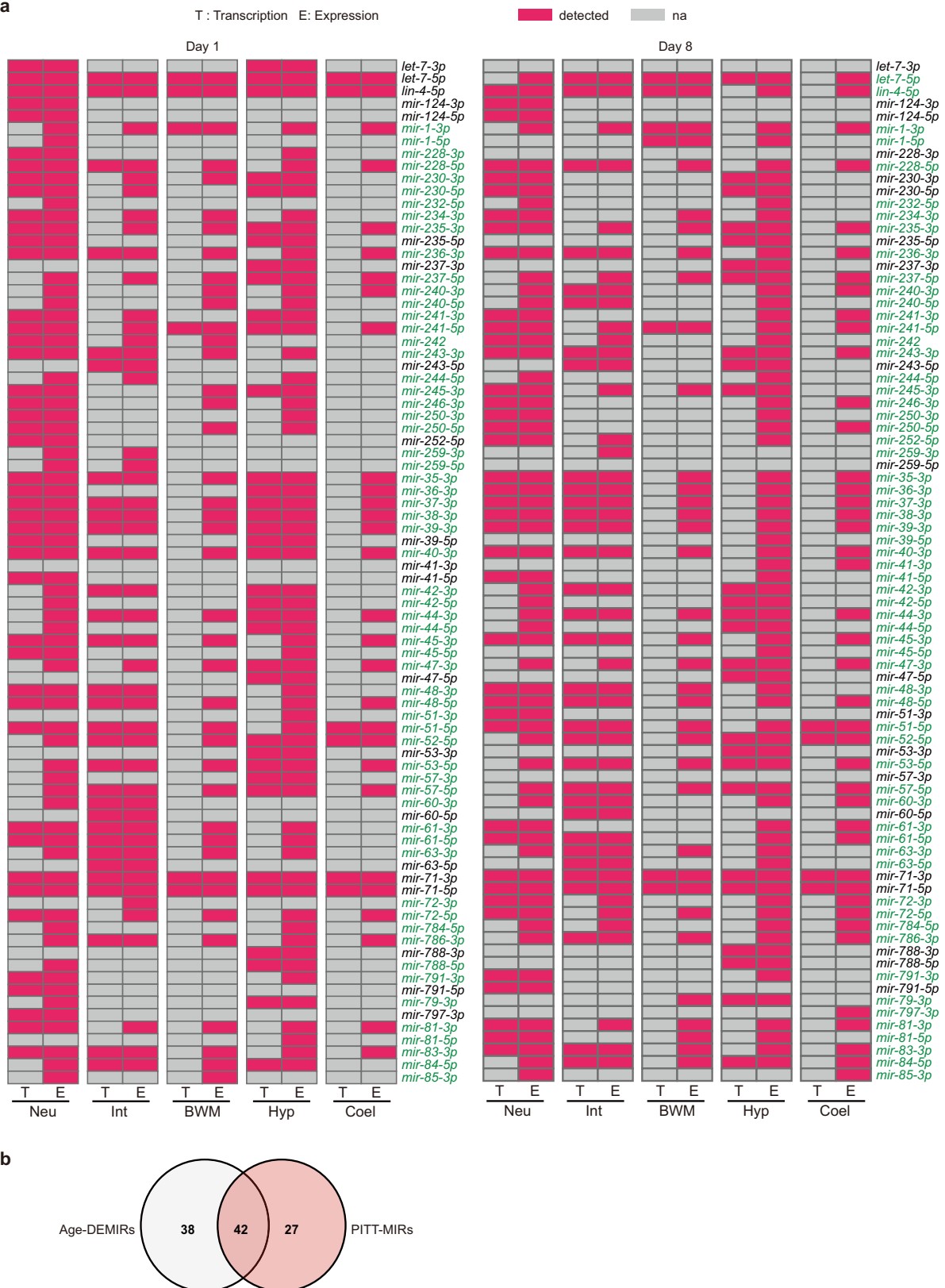

**Fig. 3 | The inconsistency between miRNA transcription and abundance in young and aged worm tissues. a** A diagram showing whether the indicated miR-NAs were detected by *PmiR::GFP* reporters (transcription, T) or miRNA-Seq (expression, E) in the examined tissues. Note that miRNAs (PITT-miRs) highlighted in green were detected by miRNA-Seq but not by *PmiR::GFP* in any of the five examined tissues. *Neu* neuron, *Int* intestine, *BWM* body wall muscle, *Hyp* hypodermis, *Coel* coelomocyte. **b** Age-DEMIRs and PITT-miRs are highly overlapping.

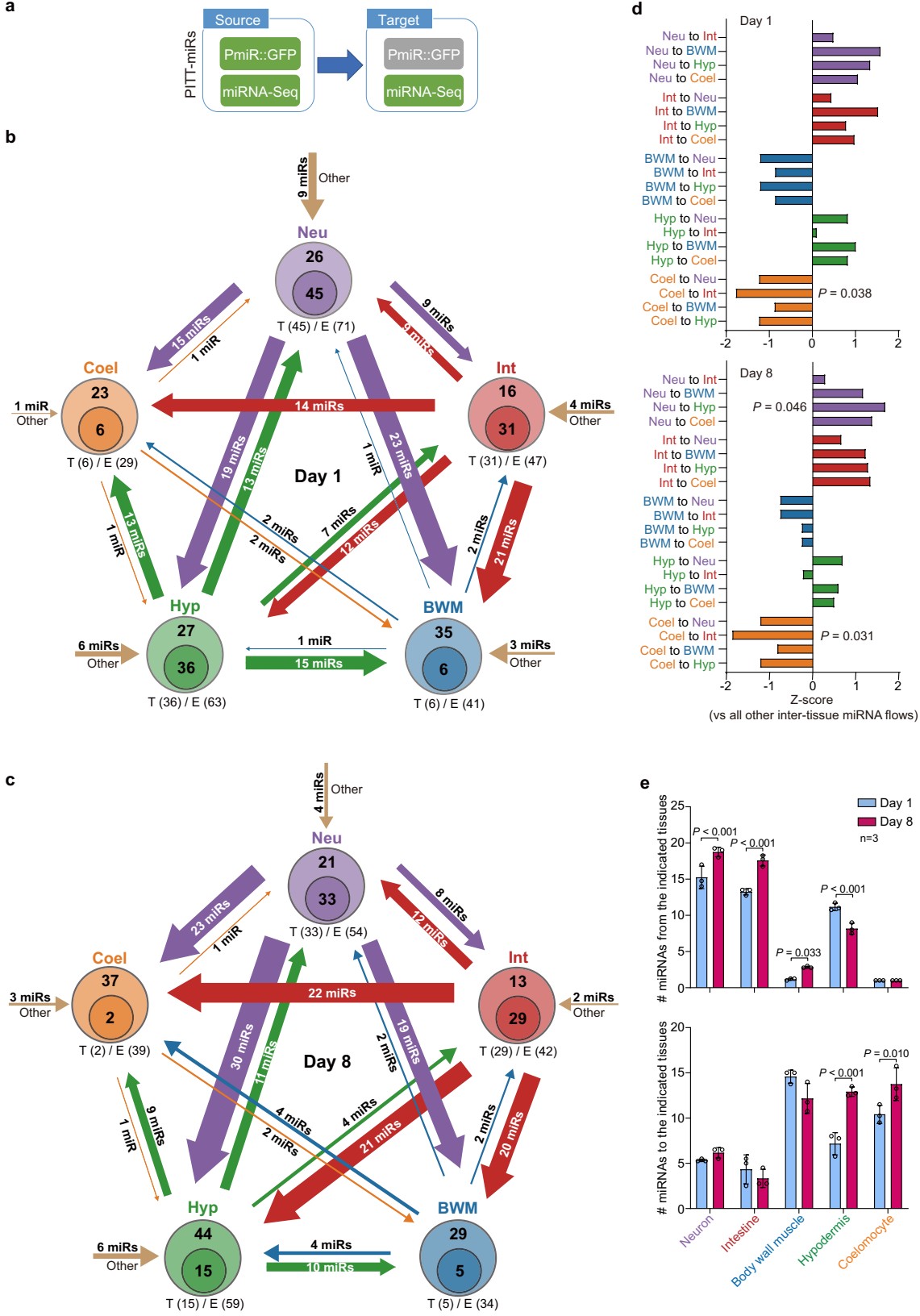

## Ageing regulates the selective secretion of miRNAs

Extracellular vesicles (EVs) transport secreted miRNAs from tissue to tissue[34]. To confirm the inter-tissue miRNA trafficking in worms and its regulation by ageing, we examined the miRNAome in the secreted EVs from worms and corresponding worm samples at D1 and D8. We detected 55 and 113 mature miRNAs in EV and the worm, respectively (Supplementary Fig. 4a–c and Supplementary Data 3). Of the 69 PITT-miRs, 25 were detected in EV as mature miRNAs (Fig. 5a, b, Supplementary Data 3). The mature forms of the rest 44 PITT-miRs were undetected in EVs, potentially due to their low expression in the worm.

**Fig. 4 | The map of inter-tissue miRNA trafficking among worm tissues. a** A diagram showing how a PITT-miR was considered to be transported from the source tissue (detected by both miRNA-Seq and *PmiR::GFP*) to the target tissue (detected by only miRNA-Seq). Green: detected. Grey: undetected. **b, c** The flow of predicted miRNA transport across indicated tissues at day 1 (**b**) and day 8 (**c**) of adulthood. Arrow thickness represents the number of transported miRNAs in this direction. The Venn diagrams for the indicated tissues show the number of PITT-miRs detected by *PmiR::GFP* (T) and miRNA-Seq (E). Note that some PITT-miRs could be transported from other tissues than the five examined ones (brown

arrows, Other). *Neu* neuron, *Int* intestine, *BWM* body wall muscle, *Hyp* hypodermis, *Coel* coelomocyte. **d** A comparison of the predicted inter-tissue miRNA flows in young and aged worms. Z-score measures the over- or under-representation of indicated miRNA flows. Two-sided permutation test. **e** The number of predicted miRNAs transported from or to the indicated tissues at day 1 and day 8 of adulthood. Two-way ANOVA, correct for multiple comparison with Tukey test. $n = 3$: three independent replicates of miRNA-Seq on indicated worm tissues. Error bars: SD. Source data of (**d**) and (**e**) are provided as a Source Data file.

---

Indeed, the detected PITT-miRs had a much higher expression in corresponding worm samples than the undetected ones (Supplementary Fig. 4d). Therefore, the PITT-miRs are very likely to be transported across tissues. In addition to these miRNAs, we found another 38 mature miRNAs in EVs (Fig. 5a, b, Supplementary Data 3). So, other miRNAs than the 69 PITT-miRs could be involved in inter-tissue miRNA trafficking.

Similar to tissue-specific miRNAomes (Fig. 1), the abundance of EV miRNAs also changed with age, with 26 upregulated and 18 down-regulated in the EVs from aged worms (Fig. 5a, b, Supplementary Data 3). We compared the abundance of each miRNA in EV with its corresponding level in the whole worm to check whether selective sorting regulates tissue-tissue miRNA trafficking. The abundance of a miRNA in the whole worm reflects its overall level[20]. If selective sorting is involved in miRNA secretion, the level of a miRNA in EVs and the whole worm should be different. Indeed, in both young and aged worm samples, the relative abundance of miRNA in EVs is not consistent with that in the whole worm. The level of this inconsistency also varies for each miRNA (Fig. 5b). These results indicate that selective sorting could control miRNA levels in EVs.

We further checked the miRNA EV loading ratio (the miRNA abundance in EVs versus that in the whole worm) in young and aged worms. Each miRNA exhibited a characteristic EV loading ratio (Fig. 5c), showing that worm miRNAs undergo selective sorting before secretion. Based on EV loading ratios, ageing promoted the secretion of 16 (30.2%) miRNAs, whereas suppressed the secretion of 17 (32.1%) miRNAs by more than 2-fold (Fig. 5c).

## MiRNAs regulate ageing across tissues

MiRNAs are signalling molecules that suppress the expression of their target genes[35]. To search for predicted miRNA targets in individual tissues, we examined the anti-correlation of age-dependent changes in miRNAs and mRNAs from our transcriptomic datasets, respectively in this study and a previous report[4] (Fig. 1), and then the alignment of miRNAs seed sequences with mRNAs 3′-UTRs using TargetScan[36]. Genes passing both analyses were considered as miRNA target genes (Fig. 6a). As expected, a large group of genes (2312) were to be regulated by the detected miRNAs in the five examined tissues during ageing (Fig. 6b and Supplementary Data 4), with many miRNAs controlling multiple targets and many genes targeted by multiple miRNAs.

The miRNA targets in worm tissues at D1 and D8 are highly similar (Supplementary Fig. 5), suggesting that the miRNA-mediated regulation of ageing is through the change in miRNA expression. We next focused on the targets of Age-DEMIRs to explore the function of the miRNAs in ageing. Across the five examined tissues, 1036 down-regulated and 1255 upregulated genes during ageing were predicted to be controlled by Age-DEMIRs (Fig. 6c and Supplementary Data 5). 9.2% of these targets have been known to regulate ageing, as indicated by GenAge[37] (Fig. 6c). WormCat analysis predicted that the targets of Age-DEMIRs control a broad spectrum of known age-related biological processes (Age-BPs)[38] (Fig. 6d and Supplementary Data 6). For example, 'metabolism' and 'proteolysis' could be regulated in multiple tissues by Age-DEMIRs[39]. Notably, various development-related gene sets, such as WNT signalling, were targeted by Age-DEMIRS (Fig. 6d

and Supplementary Data 6), underscoring the interaction between development and ageing as we previously reported[32]. Consistent with our previous study on the tissue-specific mRNA transcriptomic changes in ageing[4], Age-DEMIRs could both promote ageing and be adaptive to ageing. For example, miRNAs could inhibit 'proteolysis' and induce 'ER stress response' in aged neurons (Supplementary Data 6). Moreover, the biological processes targeted by miRNAs in ageing tissues partially overlapped with those by mRNAs (Supplementary Fig. 6), implying that miRNAs could regulate ageing in parallel with ageing-related transcription factors[4].

Inter-tissue miRNA trafficking could be prevalent and regulate miRNA compositions in worm tissues (Fig. 4). We then examined the autonomous and non-autonomous miRNA regulation on gene expression in worm tissues. The 69 PITT-miRs were considered to modulate targets (PITT-miR targets) non-autonomously in the tissues without their corresponding *PmiR::GFP* signal, and all the rest miRNA-target interactions primarily in an autonomous manner. We found that non-autonomous miRNA regulation could control around 60% of all miRNA targets in examined tissues (Fig. 7a), supporting the vital role of tissue-tissue miRNA signalling.

As 42 out of the 80 Age-DEMIRs are PITT-miRs (Fig. 3b), we further examined the role of inter-tissue miRNA signalling in ageing by focusing on the miRNAs which are both Age-DEMIRs and PITT-miRs (PITT-Age-DEMIRs). As expected, PITT-Age-DEMIRs were predicted to target a considerable number of genes non-autonomously (Fig. 7b, c), consisting of 69.7% of the Age-DEMIR targets. Of note, all PITT-Age-DEMIRs were downregulated in neurons, and 89.3% of the predicted targets of downregulated Age-DEMIRs in BWM were regulated by PITT-Age-DEMIRs (Fig. 7c). WormCat analysis also identified many Age-BPs critical to tissue functions under the regulation of PITT-Age-DEMIRs, including 'proteolysis' and 'metabolism' (Fig. 7d, and Supplementary Data 6). A large fraction Age-BPs enriched in non-autonomous miRNA targets overlapped with those modulated by Age-DEMIRs (Fig. 7e). Similarly, GenAge analysis showed that 10.0% of PITT-Age-DEMIR targets regulate ageing[37] (Fig. 7b). Therefore, inter-tissue miRNA signalling could play a critical role in the ageing of worm tissues.

## The intestinal DAF-16/FOXO is controlled by miR-1 from muscle

By *PmiR::GFP* signal, *mir-1* is exclusively transcribed in muscle[29] (Fig. 8a). However, a recent study on *mir-1* in ageing discovered that its function is not restricted in muscle, suggesting that it could act systematically[12]. In agreement with this speculation, we found that *mir-1-3p* (hereafter referred to as miR-1) could be involved in the inter-tissue miRNA signalling network. Among the examined tissues, it was transcribed only in BWM but detected in all five tissues and EV (Figs. 3a and 5b). For confirmation, we examined the level of miR-1 in the intestinal cells of WT worms, *mir-1(-)* mutants, and *mir-1*(-) mutants with an extrachromosomal array expressing miR-1 specifically in BWM (Supplementary Fig. 7a). As expected, miR-1 was detected in the intestine of WT worms but not that of *mir-1(-)* mutants (Supplementary Fig. 7b). Furthermore, expressing miR-1 in BWM restored miR-1 in the intestine of *mir-1(-)* mutants (Supplementary Fig. 7b). Therefore, miR-1 is transported from BWM into the intestine, as suggested by our datasets.

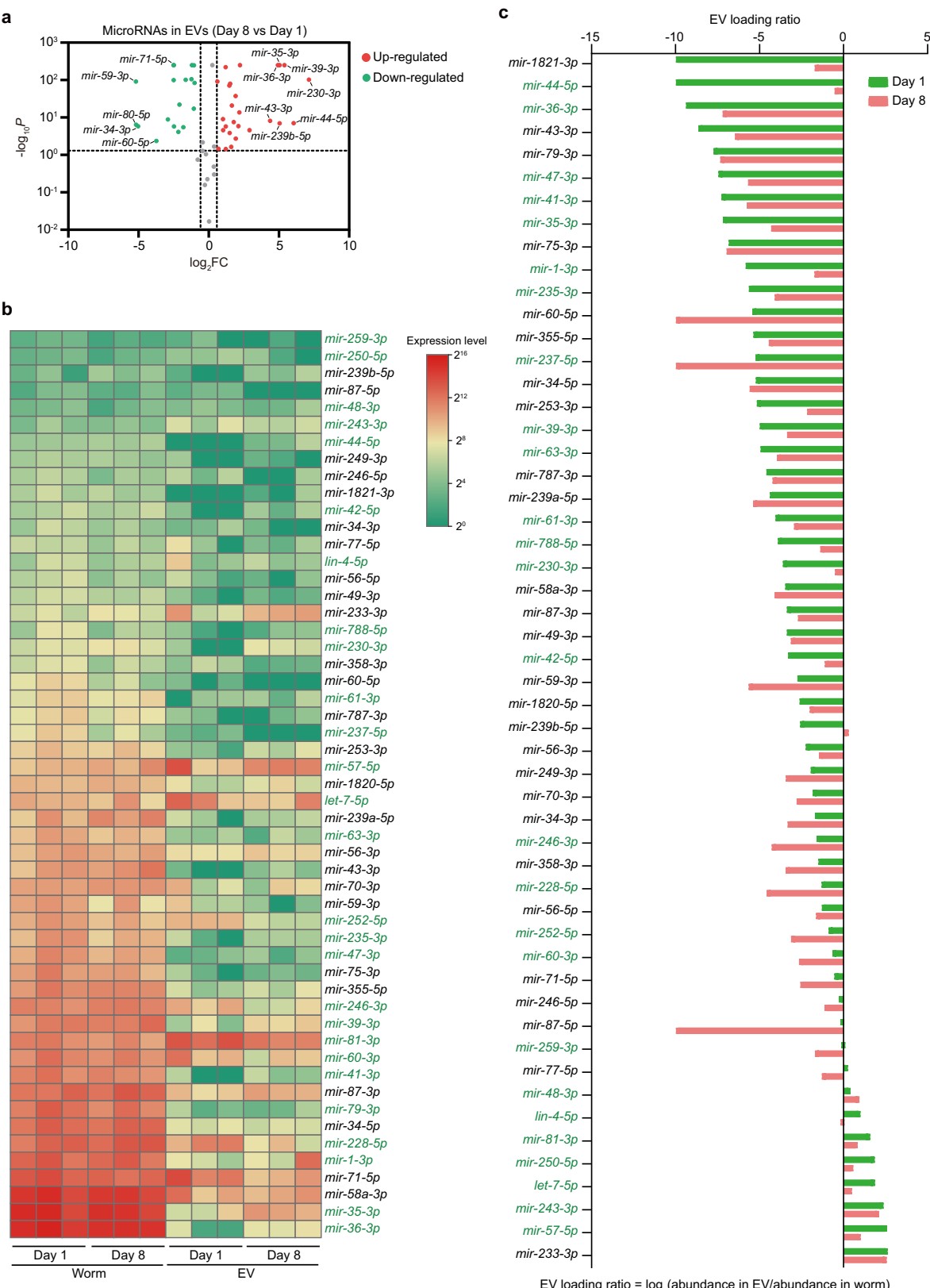

**Fig. 5 | Ageing controls the selective secretion of miRNAs. a** Ageing regulates miRNA levels in the extracellular vesicles (EVs) of worms. MiRNAs with a *P* value smaller than 0.05 and a fold of change (FC) bigger than 1.5 were considered significantly changed during ageing. Two-sided *t*-test without adjustments for multiple comparisons. **b** The relative abundance of the indicated miRNAs in the whole worm (worm) and EV from samples collected on day 1 and day 8 of adulthood. Three biological replicates were shown. **c** EV loading ratio of the indicated miRNAs at day 1 and day 8 of adulthood. PITT-miRs detected in EV are highlighted in green in b and c.

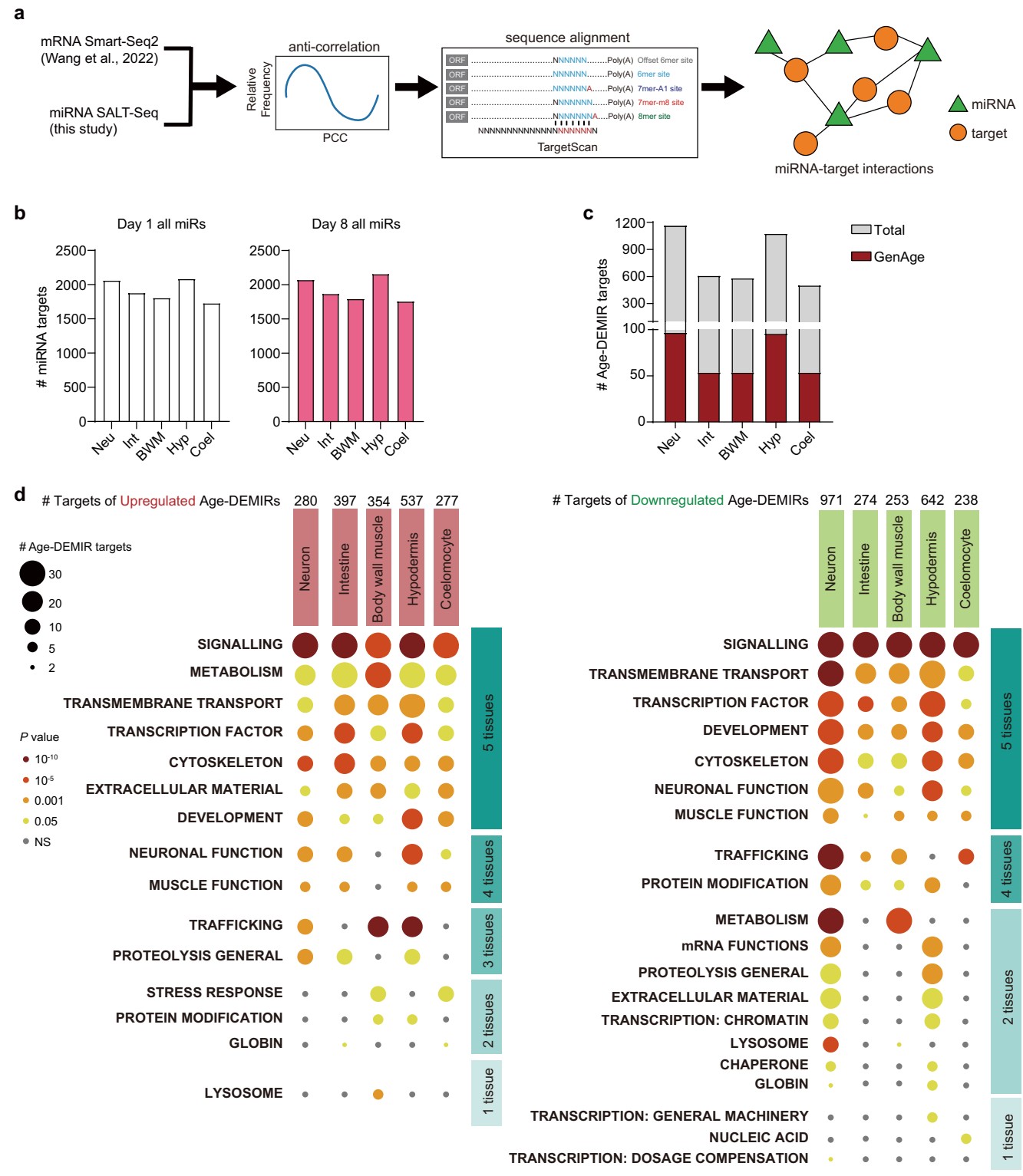

**Fig. 6 | MiRNAs regulate ageing in worm tissues. a** The flowchart depicting the identification of miRNA targets. A gene with Pearson Correlation Coefficient smaller than −0.2 and harbouring miRNA binding sites as predicted by TargetScan is considered as a miRNA target. **b** The number of miRNA targets in the indicated worm tissues at day 1 and day 8 of adulthood. **c** The number of Age-DEMIR targets in the worm tissues. Note that a remarkable fraction of Age-DEMIRs targets are involved in ageing, as annotated by GenAge. **d** The enriched gene sets from the target genes of upregulated (left) and downregulated (right) Age-DEMIRs analysed by WormCat. Only Category 1 was shown. Please see Supplementary Data 6 for detailed information on all three Categories. One-sided Fisher's exact test. *Neu* neuron, *Int* intestine, *BWM* body wall muscle, *Hyp* hypodermis, *Coel* coelomocyte.

*daf-16*/FOXO harbours a miR-1 binding site in its 3′-UTR and acts downstream of *mir-1*[12], implying it could be a miR-1 target. To test this hypothesis, we first examined the in vitro interaction between miR-1 and *daf-16* 3′-UTR. miR-1 inhibited a *daf-16*−3′-UTR luciferase reporter in HEK293T cells (Fig. 8b), confirming their interaction. We next used CRISPR/Cas9 technology to tag endogenous *daf-16* with GFP::3xFLAG at its C-terminal and examined its expression in WT worms and *mir-1(-)* mutants. By western blot, DAF-16::GFP::3xFLAG was remarkably up-

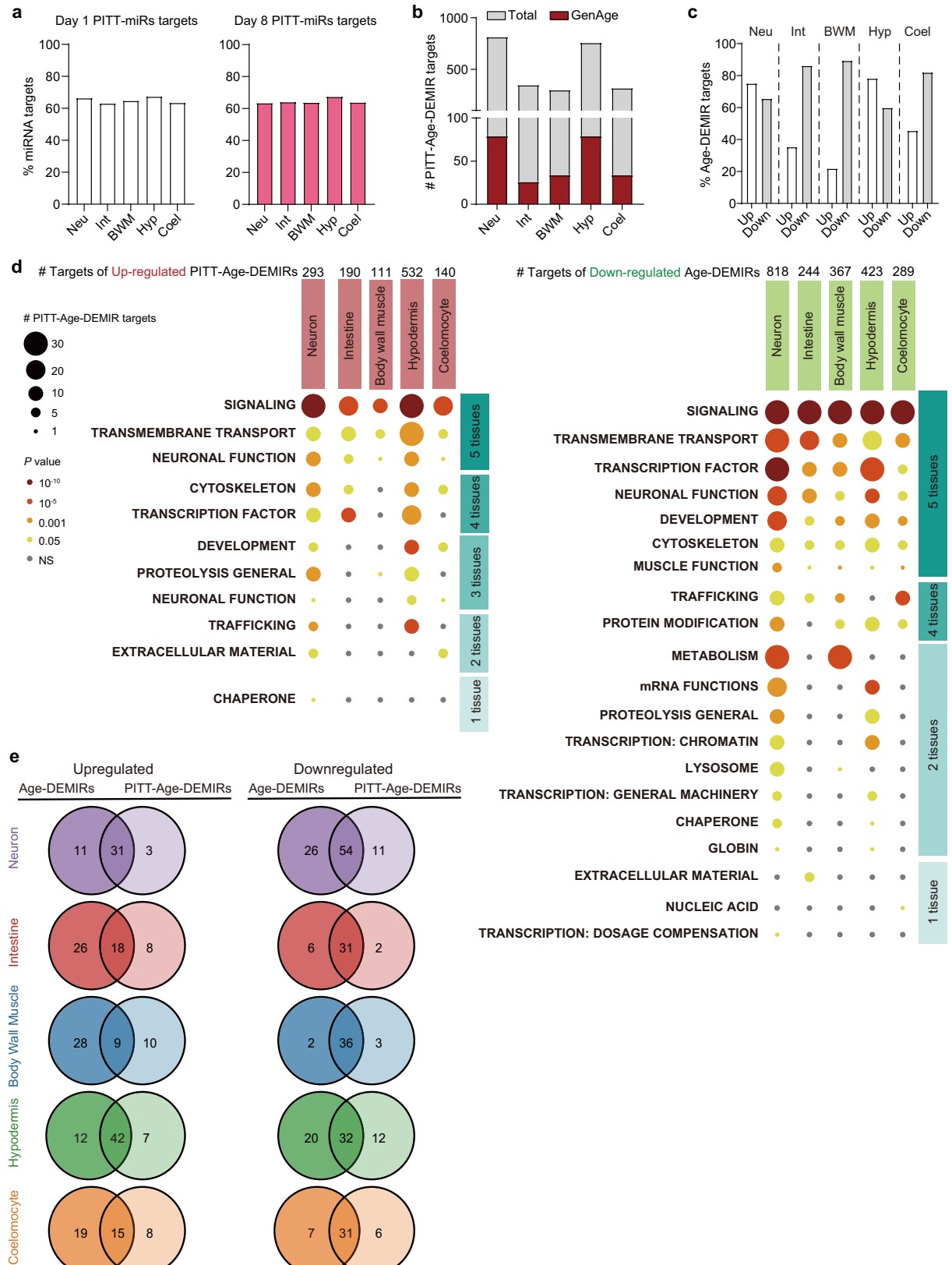

**Fig. 7 | Inter-tissue miRNA signalling regulates ageing. a** A remarkable fraction of miRNA targets are regulated by PITT-miRs from other tissues in the indicated worm tissues at day 1 and day 8 of adulthood. **b** The number of PITT-Age-DEMIR targets in the worm tissues. The number of PITT-Age-DEMIRs targets involved in ageing, as annotated by GenAge, are highlighted in red. **c** The percentage of Age-DEMIR targets regulated by PITT-Age-DEMIRs in the indicated worm tissues. **d** The enriched gene sets from the target genes of upregulated (left) and downregulated (right) PITT-Age-DEMIRs analysed by WormCat. Only Category 1 was shown. Please see Supplementary Data 6 for detailed information on all three Categories. One-sided Fisher's exact test. **e** The Venn diagrams comparing the enriched gene sets (Category 3 by WormCat) of Age-DEMIR targets and PITT-Age-DEMIR targets in indicated worm tissues. *Neu* neuron, *Int* intestine, *BWM* body wall muscle, *Hyp* hypodermis, *Coel* coelomocyte.

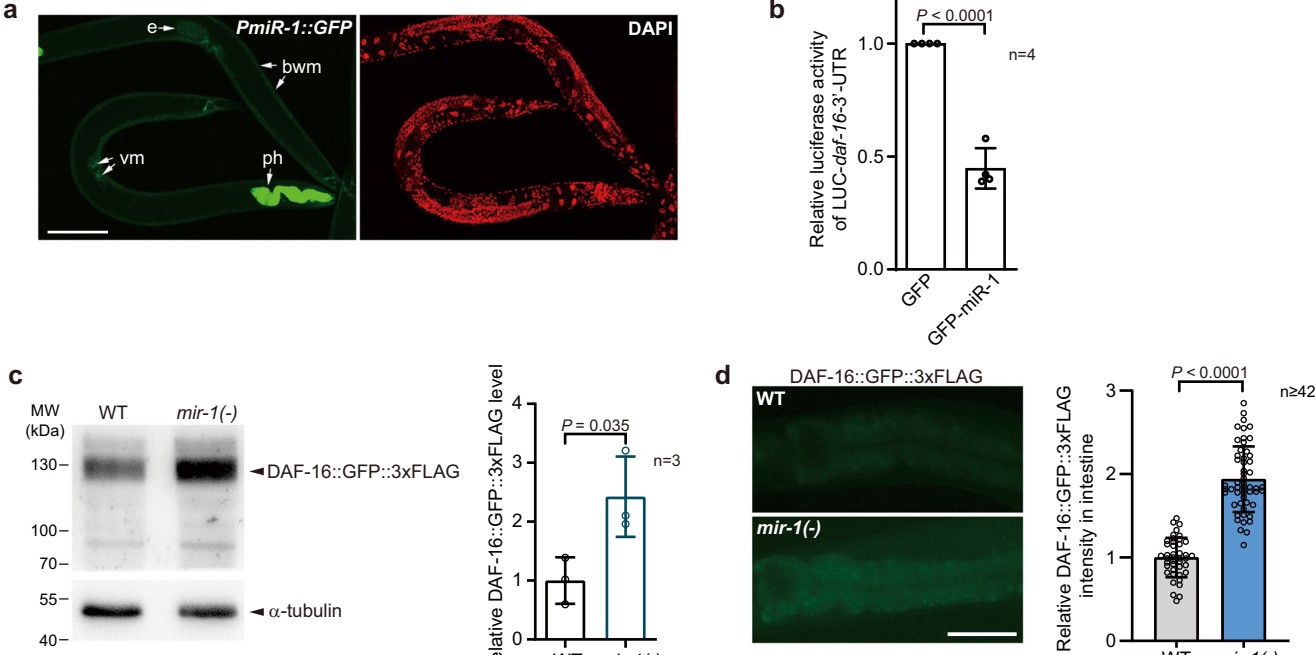

**Fig. 8 | miR-1 controls intestinal DAF-16 through inter-tissue transportation.**
**a** The pattern of miR-1 transcription across worm tissues. Note that *PmiR-1::GFP* is expressed only in worm muscles. ph: pharynx, bwm: body wall muscle, vm: vulva muscle, e: egg. Scale bar: 50 μm. A representative image from 26 examined worms was shown. **b** Overexpressing miR-1 suppresses a luciferase reporter of *daf-16* 3′-UTR in HEK293T cells. *n* = 4: four biological replicates. **c** Mutating *mir-1* increases DAF-16::GFP::3xFLAG at the worm level. α-tubulin served as the loading control. *n* = 3: three biological replicates. **d** DAF-16::GFP::3xFLAG in the intestine of the indicated strains. Scale bar: 50 μm. 42 WT worms and 54 *mir-1(-)* mutants were examined. Unpaired *t*-test (two-sided) (**b**–**d**). Error bars: SD. Source data are provided as a Source Data file.

regulated upon the mutation of *mir-1* (Fig. 8c). As western blot shows the protein level from the whole worm, the miR-1-controlled increase of DAF-16 is unlikely to occur only in muscles. Indeed, when further examining the GFP signal in worms, we found a similar increase of DAF-16::GFP::3xFLAG in the intestine of *mir-1(-)* mutants (Fig. 8d). Therefore, miR-1 from muscle suppressed DAF-16 in the intestine.

*daf-16*/FOXO is the pivotal TF in the longevity pathway of insulin/IGF-1 signalling (IIS)[40]. Our RT-qPCR analysis found that miR-1 is suppressed in *daf-2(-)* mutants in a *daf-16*-dependent manner (Supplementary Fig. 7c). Hence, miR-1 could mediate an IIS signal flow from muscle to the intestine.

## Discussion

MiRNAs are critical regulators of ageing[11,13,19,32,41]. Whereas ageing induces systematic changes across tissues, previous ageing studies on individual miRNA or miRNAome were at the organismal level or focused on a specific tissue, lacking a global view of miRNA in the ageing of various tissues[13,21,42–45]. Thus, studies on miRNA in the ageing and co-ageing of distinct tissues have been rather limited.

*C. elegans* is an excellent model organism in a wealth of biological research, including miRNA and ageing[17,19,46]. In particular, *C. elegans* has a variety of tissues similar to higher organisms in structure, gene expression, and ageing phenotypes[46,47]. However, due to the tiny size of *C. elegans*, dissecting its tissues to profile tissue-specific miRNAomes is challenging. Several approaches were reported to overcome this technical issue. One is to selectively methylate miRNAs by expressing an Arabidopsis methyltransferase, HEN1, in the worm tissue of interest and clone the methylated miRNAs for sequencing[25]. One is to immunoprecipitate Argonaute (AGO) protein expressed tissue-specifically and profile the associated miRNAs[26]. The last one is through FACS of the nucleus which are labelled using tissue-specific fluorescent proteins[27]. None of the methods directly profile miRNAs in the worm cells. The first two highly depend on the efficiency of HEN1-mediated methylation or the co-immunoprecipitation with AGO[25,26]. These extra steps could potentially affect their detection sensitivity. Moreover, since these two methods sample the whole worm, miRNAs from larger tissues (e.g., intestine) could be much easier to be detected than those from smaller ones (e.g., coelomocytes). As for the last one, it discards cytoplasm, focusing on a fraction of total miRNAs[27].

In this study, we physically isolated cells from five major worm tissues to profile tissue-specific miRNAomic changes during worm ageing. Because the cells in worm tissues are notably homogeneous, the isolated cells are highly representative of corresponding tissues, although we cannot wholly exclude the variations in the same type of cells. As reported in our previous study[4], quality control with highly-expressed tissue-specific genes shows that our approach does not suffer from cross-contamination of debris from other tissues. Besides, we executed miRNA-Seq directly on isolated cells without additional treatment, thereby avoiding the potential drawbacks of previous methods.

Our research identified numerous miRNAs in worm tissues not reported in the previous three reports[25–27] (Supplementary Fig. 1 and Supplementary Data 2). For example, we detected miR-1-3p in all examined tissues, whereas this miRNA was only found in BWM but not in neurons or intestine by tissue-specific AGO immunoprecipitates[26]. As we speculated, the level of miR-1-3p in neurons or intestine is significantly lower than that in BWM, as evidenced by our datasets. Moreover, we reported the miRNAs from a tiny but vital worm tissue, coelomocytes, which consist of only six cells. Unlike previous reports, which analysed larvae or worms of mixed stages[25–27], we focused on miRNAomic information of adult worm tissues at different ages. Our datasets thus provide a valuable resource for comparative ageing studies of miRNA across tissues.

As expected, miRNAs are essential regulators in worm tissues, predicted to control thousands of protein-coding genes (Fig. 6, Supplementary Data 4). Accordingly, we found that Age-DEMIRs could control many critical biological processes in ageing, including

metabolism, proteolysis, and stress response[48] (Fig. 6, Supplementary Data 6). We previously reported ageing-related biological processes in individual worm tissues based on the differentially expressed mRNA genes (Age-DEGs)[4]. Only part of the Age-DEMIRs was predicted to target biological processes overlaps with the Age-DEG-controlled ones (Supplementary Fig. 6), implying that miRNAs control tissues ageing complementary with the age-dependent transcriptomic changes of mRNAs. Of note, these potential age-related miRNA targets are reported to be both anti- and pro-ageing[49], suggesting that the miRNA-based signalling could both induce and be adaptive to ageing.

In addition to their intracellular functions, recent studies showed that miRNAs transcribed in one cell can be transported into another and act as messengers in intercellular signalling[9,14,50]. Nevertheless, a global view of the inter-tissue miRNA signalling network at the organism level is yet to be established.

We detected 69 miRNAs (PITT-miRs) in the miRNAome but not by *PmiR::GFP* in at least one analysed tissue (Fig. 3). Such a discrepancy is unlikely to be from miRNA processing or degradation, as these two processes do not regulate *PmiR::GFP* and their interference could only cause a miRNA to be detected by *PmiR::GFP* but not by miRNA-Seq[51]. Instead, the inconsistent transcription and abundance we observed strongly suggest inter-tissue miRNA transport. The difference between *PmiR::GFP* and miRNA-Seq results could also come from the different sensitivities of the two methods. However, many PITT-miRs were detected in purified worm extracellular vesicles (EVs), which deliver miRNA from cell to cell[34] (Fig. 5), supporting our speculation that PITT-miRs are transported across tissues. Some PITT-miRs were undetected in EV. This could be either due to their low expression or because they are transported from tissue to tissue in a non-EV manner (e.g., apoptotic body or non-vesicle form)[52,53]. Therefore, our datasets strongly suggest that inter-tissue miRNA trafficking is prevalent among worm tissues. Moreover, the families of PITT-miRs have a more ancient origin than all detected miRNAs (Supplementary Data 1), implying that miRNA could function in cell-to-cell communication early in evolution.

Based on PITT-miRs, we mapped the miRNA trafficking network across the five examined worm tissues (Fig. 4). MiRNA flow is present between all tissue pairs, underscoring the prevalence of inter-tissue miRNA trafficking. MiRNA flow also exhibits strong tissue specificities, with their sizes and compositions varying by the source and recipient tissues. We postulate that this is mainly due to different miRNA transcription across tissues, rather than a tissue-specific preference to uptake EVs with particular miRNAs. As shown by previous reports and our own *PmiR::GFP* analysis[29] (Fig. 3), miRNA transcription exhibits strong tissue-specificities, defining the various tissue-tissue miRNA flow at the starting points. The tissue-specificity could reflect a general flow of inter-tissue miRNA signalling. Neuron, intestine, and hypodermis function more as signalling towers, whereas BWM and coelomocytes act more as receivers. However, it is worth noting that the inter-tissue miRNA signalling is not unidirectional but reciprocal. Additionally, our miRNA-Seq found some PITT-miRs without *PmiR::GFP* signal in the examined tissues, implying that these miRNAs could be taken in from other worm tissues. PITT-miRs control approximately 60% of miRNA targets in each examined tissue (Fig. 7), implying a remarkable role of the tissue-to-tissue miRNA trafficking network in *C. elegans* biology. Taken together, our datasets suggest that all tissues work in a well-coordinated manner via inter-tissue miRNA signalling.

In addition to PITT-miRs, we detected 38 miRNAs in worm EVs (Fig. 5). A miRNA with similar transcription and abundance could also be delivered inter-cellularly as long as its secretion and absorption are balanced. Therefore, inter-tissue miRNA trafficking could involve more miRNAs than PITT-miRs. A worm miRNA is then highly likely to function beyond the tissue of its transcription, highlighting the necessity to pursue the non-autonomous functions of miRNAs in future studies. It will also be important to consider miRNA when investigating intercellular signalling.

The inter-tissue miRNA network is regulated by ageing (Fig. 4 and Supplementary Fig. 3). The age-dependent change in miRNA transcription controls tissue-tissue miRNA signalling, as expected. Besides, we identified miRNA sorting as another essential regulation of the inter-tissue miRNA network in ageing (Fig. 5). MiRNAs are selectively sorted into EVs for secretion[54,55]. We found that the miRNAomes in worm EV and the whole worm change differently during ageing (Fig. 5, Supplementary Data 3), showing the importance of miRNA sorting in the composition of circulating miRNAs. The mechanism underlying miRNA sorting remains to be determined despite recent advances[55–59]. It is intriguing to explore the machinery of selective miRNA secretion and its role in ageing in the future. Moreover, the age-dependent structural change of worm tissues could also alter the inter-tissue miRNA network. For example, our datasets suggest an upregulation of miRNA trafficking from the intestine to other tissues in aged worms, possibly due to the increased intestinal permeability during ageing. We are unable to distinguish the EVs from different tissues in this study. To further understand the inter-tissue miRNA signalling, it will be critical to dissect tissue-specific EVs in the future.

Secreted miRNAs have been proposed as biomarkers of ageing[15,16,60,61]. The inter-tissue miRNA network we discovered indicates that they could also deliver ageing signals from tissue to tissue. In line with this speculation, miRNAs in worm EVs have been proposed to regulate ageing[62]. Similarly, we previously found that *mir-83* transported across tissues coordinates the age-dependent dysregulation of macroautophagy in distinct tissues[13]. Our datasets further suggest that a significant fraction of other inter-tissue miRNA targets is involved in ageing (Fig. 7, Supplementary Data 5). Together with previous reports[63–65], the extensive reciprocal miRNA flows across tissues also support a "consensus" mode that ageing is controlled based on signals from various tissues.

To validate the miRNA signalling network identified from our datasets, we examined miR-1 (Fig. 8 and Supplementary Fig. 7), which was recently identified as a critical regulator of lysosomes during ageing[12]. Although the transcription of miR-1 is restricted in muscle, mutating *mir-1* alters lysosome-related phenotypes in multiple other tissues[12,29]. Indeed, our inter-tissue miRNA network map, EV miR-NAome, and RT-qPCR of isolated worm tissues indicate that miR-1 is transported across tissues. Following the lead that *daf-16* is up-regulated in *mir-1(-)* mutants[12], we further discovered that miR-1 from BWM suppressed DAF-16 expression in the intestine. Moreover, miR-1 itself is inhibited by *daf-16* in *daf-2(-)* mutants. DAF-16/FOXO is the pivotal transcription factor in insulin/IGF-1 signalling (IIS)[40]. Therefore, our findings suggest a miRNA-mediated non-autonomous regulation of IIS. When IIS is disrupted (e.g., *daf-2*/IR mutation), this circuit is repressed by DAF-16 in muscle with a subsequent decrease of miR-1, enhancing both muscular and intestinal DAF-16 activity and consequently regulating lysosomal function.

In summary, our study provides tissue-specific miRNAomic information and reveals a complex inter-tissue signalling network of miRNA in the ageing of *C. elegans*. Our datasets are thus a precious resource to investigate the autonomous and non-autonomous miRNA-based regulation of ageing amongst tissues. Moreover, many Age-DEMIRs identified in this study are well-conserved and present in human serum[16]. Following this study, it will be interesting to investigate their roles in the ageing of higher organisms. To better understand the regulation of miRNA in ageing, it will also be necessary to explore other facets of miRNA biology in ageing, including miRNA biogenesis, miRNA isoforms, loading into the RNA-induced silencing complex, and degradation[26,30,31,51].

## Methods

### *C. elegans* strains and culture

Worm strains used in this study are listed in Supplementary Data 8. Some strains were provided by CGC, which is funded by the NIH Office

of Research Infrastructure Programs (P40 OD010440). All assayed worms were cultured with the standard technique at 20°C[66].

For small RNA and mRNA sequencing of isolated cells from specific worm tissues, ~2000 synchronised WT worms of day 1 adulthood were collected. For day 8 samples, ~14,000 synchronised worms were raised on regular NGM until L4 and subsequently transferred to FUdR-containing plates supplemented with 25 µM FUdR (Adamas)[67,68]. Worms were washed three times with M9 to deplete the remaining bacteria and then transferred onto fresh NGM plates containing FUdR every other day from L4 to day 8 of adulthood.

For miRNA-Seq of whole worm and worm EVs, synchronised worms were grown in liquid culture medium (S medium with 25 mg OP50 per millilitre) with 3 worms per microliter until day 1 of adulthood to get young worms. L4 worms grown in liquid were transferred into a liquid culture medium supplemented with 25 µM FUdR, with 3 worms per microliter to get aged worms. Worms were transferred to a fresh FUdR-supplemented liquid culture medium every other day until day 8 of adulthood.

## Plasmid construction

sgRNA for *daf-16* was designed by Zhang lab's CRISPR design tool at http://crispr.mit.edu and inserted into pDD162 (a gift from Bob Goldstein, Addgene #47549). The homology recombination template for *daf-16* was constructed by cloning the ~0.6 kb of 5′ and 3′ homology arms into pDD282 or pDD284 plasmids (gifts from Bob Goldstein, Addgene # 66823) using Gibson Assembly® Cloning Kit (NEB). The target site in the template was modified with synonymous mutations.

Plasmids used in luciferase assays were prepared as previously described[11]. For plasmids expressing mature *cel-mir-1-3p*, the miRNA stem-loop and ~200 bp of the flanking sequence were amplified from genomic DNA and inserted into the 3′-UTR of GFP in pEGFP-C1.

All plasmids have been submitted to BRICS (http://www.brics.ac.cn/plasmid/template/article/about.html).

## Transgene

GFP::3xFLAG was knocked in at the C-terminal of *daf-16* as reported[69].

## Isolation of worm tissues

Worm tissues were isolated as reported[4]. In brief, worms with tissue-specific fluorescent markers were incubated with SDS-DTT and proteolysis with mechanical disruption[13,47]. A total of 20–40 fluorescent cells from the intestine, body wall muscle, hypodermis, and coelomocyte were manually picked by an Eppendorf TransferMan 4r mounted on an Olympus IX73 microscope for each biological replicate[13]. For neurons, worm lysates were filtered with a 5-µm cell strainer. Filtered worm lysates were subjected to FACS for YFP-labelled neurons[47] (Supplementary Fig. 8). Around 8000 neurons were collected for each biological replicate. N2 worms at the corresponding age were similarly treated as the negative control of auto-fluorescence. Tissue purity was validated by RT-qPCR of tissue-specific genes before RNA library construction as reported[4]. Three biological replicates were collected for each tissue at each age.

## Extracellular vesicle preparation

The extracellular vesicles were prepared as reported[13,62]. In brief, ~200,000 worms were washed by M9 to remove excess bacteria and cultured in 60 ml M9 at 20 °C for 6 h with shaking at 180 rpm. Worms were then sedimented by gravity, and larvae were depleted by centrifugation at 4000 rpm for 5 min. Sedimented worms were collected into QIAzol (QIAGEN). The supernatant was collected and filtered by a 0.22 µm Millex-GP Syringe Filter Unit (Millipore) and then centrifuged at 25,000 *g* at 4 °C for 30 min. The supernatant was further subjected to ultracentrifuge twice at 100,000 *g* at 4 °C for 30 min. Sedimented EVs were resuspended in PBS for quality control or in QIAzol (QIAGEN)

for miRNA-Seq. Worms or EVs collected in QIAzol (QIAGEN) were subjected to miRNA preparation using miRNeasy Mini kit (QIAGEN)[11].

## Transmission electron microscope on EV

A total of 10–20 µl EVs suspended in PBS were deposited onto a carbon-coated electron microscopy grid, stained with 2% uranyl acetate, and examined using a Tecnai G2 Spirit microscope (Thermo Fisher Scientific).

## Nanoparticle track analysis

EVs in PBS were diluted 100-fold in PBS and subjected to nanoparticle tracking using a NanoSight NS300 (Malvern Instruments).

## Small RNA sequencing of isolated cells from worm tissues

Small RNA libraries were constructed as previously described[23]. The isopropanol precipitation method combined with a sedimentation aid was used to extract total RNA from worm neurons. Briefly, worm neurons isolated by FACS were collected in TRIzol (Thermo Fisher Scientific) and subsequently mixed with chloroform. The upper aqueous phase was mixed with a same volume of isopropanol, 10% volume of 3 M NaAc (pH 5.2), and 3 µl of 5 µg/µl LPA were added. The mixture was then incubated overnight at −20 °C. Precipitated RNA was collected by centrifugation, washed twice with 75% ethanol, and dissolved in RNase-free water. Other worm cells isolated by manual picking are collected in 4 µl cell lysis buffer of 0.01% RNaseOUT™ Recombinant Ribonuclease Inhibitor (Thermal Fisher Scientific) and 0.2% Triton X-100 in nuclease free water. Extracted RNA or cell lysates were incubated at 72°C for 3 min and subjected to 3′ adaptor ligation. Lambda exonuclease and 5′ Deadenylase were used in combination to remove the excess 3′ adaptor. 5′ adaptor ligation and reverse transcription reaction were followed. Pre-amplification was performed, and 1 µl of the product was used for final amplification. The amplified libraries were separated on a 6% polyacrylamide gel, and 130–160 bp DNA fragments were sliced. All libraries were sent for sequencing on an Illumina HiSeq X Ten platform or NovaSeq 6000. Critical quality control information analysed by miRTrace is shown in Supplementary Data 9[70].

## MiRNA-Seq of whole worm and worm EVs

MiRNA-Seq of whole worm and worm EVs were performed by BGI. In brief, 1 µg of total RNA prepared by miRNeasy Mini Kit (QIAGEN) was used for library preparation for each sample. Total RNA was further purified by electrophoresis on a 15% urea denaturing polyacrylamide gel. The small RNA regions corresponding to 18–30 nt were excised and recovered. Small RNAs were subsequently ligated to adenylated 3′ adaptors annealed to unique molecular identifiers, followed by the ligation of 5′ adaptors. The adaptor-ligated small RNAs were transcribed into cDNA by SuperScript II Reverse Transcriptase (Invitrogen) and PCR-amplified. The amplified libraries of 110–130 bp were purified by QIAquick Gel Extraction Kit (QIAGEN) and then sequenced using the BGISEQ-500 platform (BGI-Shenzhen).

## RNA-Seq data analysis

For tissue-specific miRNomic analysis, sequencing reads were mapped to WBcel235 by STAR (v2.6.0c)[71]. MirGeneDB 2.1 was used for miRNA annotation[24]. Mapped reads were quantified by feature counts[72] and normalised by TPM (Transcripts Per Million). Quantile normalisation was performed on $\log_2$TPM. Total genes were used to calculate the Pearson correlation coefficient and perform principal component analysis by scikit-learn[73].

For miRNA-Seq of whole worms and isolated EVs, reads were mapped to WBcel235 by miRDeep2[74]. Differentially expressed miRNAs were identified using DEGseq based on unique molecular identifier (UMI)[75]. A minimum UMI sum of 10 in 3 replicates was set as the threshold of expression.

MiRNAs with more than five reads were defined as expressed. Differential expression of miRNAs was analysed by *t*-test (*P* value < 0.05 and fold-change >1.5 or <0.67) after Box-Cox transformation. MiRNA targets were identified by TargetScanWorm (Release 6.2) and Pearson Correlation Coefficient smaller than −0.2.

## Homology annotation
Human homology of worm miRNAs was annotated following Ibanez-Ventoso, C. et al.[76].

## Pathway analysis
Pathway analysis was performed using WormCat 2.0[38]. Terms were considered as significant when the *P* value was smaller than 0.05.

## Cell culture and transfection
HEK293T cells were from ATCC and maintained in DMEM (Thermo Fisher, Cat# C11995500CP) medium supplemented with 10% fetal bovine serum (Thermo Fisher, Cat# 10099141) at 37 °C, 5% $CO_2$. Cells were authenticated by morphology and tested for mycoplasma contamination before experiments. Transfections were performed according to the manufacturer's instructions with Lipofectamine 3000 (Invitrogen).

## Luciferase assay
Cells were collected and examined for luciferase activity 48 h post-transfection by Dual-Luciferase® Reporter Assay System (Promega) as the manufacturer instructed.

## Microscopy
Worms were anaesthetised using 2 μM levamisole and mounted on 5% agar pads. Images were collected using an Olympus BX53 or a Leica TCS SP8 X microscope. The GFP intensity of DAF-16::GFP::3xFLAG was measured by ImageJ (1.53t). Background signals were subtracted as reported[77].

## Western blot
Worms were harvested in M9. After three rounds of freezing and thaw, worms were lysed by 4x SDS loading buffer (Takara, Cat#9173). Proteins were separated by reducing SDS-PAGE and transferred to nitrocellulose membranes. Membranes were then blotted with primary antibodies against GFP (1:2000, Santa Cruz, Cat# sc-9996) or α-tubulin (1:5000, Sigma, Cat# T5168), and HRP-conjugated secondary antibodies against mouse IgG (1:5000, TermoFisher Scientific, G-21040). Signals of western blotting were captured by a Tanon™ 5200 Chemiluminescent Imaging System and measured using ImageJ[78].

## RT-qPCR
RT-qPCR from worms were performed as reported[11]. Young adult animals without eggs were examined unless otherwise noted. Following miRNA preparation by miRNeasy Mini kit (QIAGEN), TaqMan MicroRNA Reverse Transcription kit (Applied Biosystems) was used for cDNA preparation. qPCR was performed with Power SYBR Green master mix (Applied Biosystems) on a ViiA 7 Real-Time PCR System or a 7900HT Fast Real Time PCR System (Applied Biosystems). Three or four technical replicates were performed in each reaction. Relative transcript levels were calculated by the ddCt method, and snRNA sn2841 was used as the reference gene. For RT-qPCR from isolated worm cells, cell lysates were aliquoted into two for mRNA and microRNA analysis, respectively. Reverse transcription and qPCR were performed as reported[13]. qPCR primers were listed in Supplementary Data 8.

## Statistics & reproducibility
Results are presented as Mean ± SD unless otherwise noted. To analyse the significance of over-represented or under-represented inter-tissue miRNA transport by permutation test, the edges were randomised for 1000 times. The randomly sampled numbers of miRNAs were transformed to a normal distribution by Yeo-Johnson transformation. The *P*-value was estimated by the cumulative distribution function and survival function in SciPy. Other statistical tests were performed as indicated using GraphPad Prism 6.01 (GraphPad software). Venn diagrams and Upset plots were generated using TBtools[79]. No data were excluded from the analyses. Detailed statistics for experiments are listed in Supplementary Data 7.

## Reporting summary
Further information on research design is available in the Nature Portfolio Reporting Summary linked to this article.

## Data availability
The miRNA-Seq data generated in this study have been deposited in the SRA database. Small RNA-Seq data of isolated tissues from worms at D1 and D8 of adulthood were deposited under accession codes SRX16013599-SRX16013628 [https://www.ncbi.nlm.nih.gov/sra/?term=PRJNA854735]. Small RNA-Seq data of extracellular vesicles and worms at D1 and D8 of adulthood were deposited under accession codes SRX17016274-SRX17016279 [https://www.ncbi.nlm.nih.gov/sra/?term=PRJNA868152]. RT-qPCR primers are provided in Figshare: https://doi.org/10.6084/m9.figshare.22799864. Raw data files of images are available from the corresponding author on reasonable request due to the size limitation of public data deposition service. The rest of the data generated in this study are provided in Supplementary Information and Source Data files. Source data are provided with this paper.

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

## Acknowledgements

The authors thank the institutional core facilities for cell biology and molecular biology for instrumental and technical support, and Ms Wen Xiao for her assistance in miRNA-Seq analysis. The depiction in Fig. 1a was created with BioRender.com. This research was supported by the Strategic Priority Research Program of the Chinese Academy of Sciences, Grant No. XDB19000000 to Y.S., and grants from China Ministry of Science and Technology (2020YFA0804000), National Natural Science Foundation of China (92049302, 32088101, 91749205), and Shanghai Municipal Science and Technology Major Project (2017SHZDZX01) to J.J.H. The illustration in Fig. 1a is modified from J. J. Froehlich's artwork under CC BY-SA 4.0.

## Author contributions

X.W., L.W., J.J.H., and Y.S. (Yidong Shen) conceived the project and designed the experiments. X.W., Z.H., and Y.Z. collected worm tissues, worms, and EVs for sequencing, performed worm-related experiments and analysed the data with the assistance of Y.S. Y.S. (Yuanyuan Song) constructed RNA libraries under the supervision of H.Z., Y.L., and L.W. Q.J. and X.W. performed bioinformatics analysis under the supervision of J.J.H. and Y.S. (Yidong Shen). Z.H., Y.S. (Yidong Shen), and performed mir-1 related assays and analysed the data with A.A. Y.C., N.T., and X.W. analysed PmiR::GFP reporters in worm tissues. Y.S. (Yidong Shen) wrote the manuscript. All authors contributed to manuscript editing.

## Competing interests

The authors declare no competing interests.
