## [Peer Review File · Nature Communications]

Tissue-specific profiling of age-dependent miRNAomic changes in *Caenorhabditis elegans*REVIEWER COMMENTS

Reviewer #1 (Remarks to the Author):

REVIEW NCOMMS-23-20619

The manuscript: “Tissue-specific profiling of age-dependent miRNAomic changes in *Caenorhabditis elegans*” is conceptually very interesting. For the main claims of this manuscript - inter-tissue trafficking as measured by the lack of transcription but the presence of microRNAs - however, I see substantial shortcomings. Many aspects of the study are too briefly explained and conclusions too bold in the light of the presented analyses /results.

Use of miRBase:

For years, a major concern in microRNA research has been the quality of the online repository miRBase (1–14) with estimates of 2/3 false-positive entries. Thus, the database contains many false-positives and not only microRNAs. These are for instance numerous tRNA, rRNA or other fragments and transcriptional noise but also incorrectly annotated bona fide miRNAs that strongly influence interpretations of data. In addition to the false positives, miRBase annotations are often imprecise and have varying precursor annotation forms (with or without flanking regions of varying lengths) and very often not both arms are annotated, 3' ends are incorrect, and in a few cases even 5' are not correctly annotated which substantially affects target predictions, and more so, in this case here when needing accurately defined Drosha and Dicer cutting sites. Further it is using an outdated nomenclature which is inconsistent and leads to problems of not-summarizing inconsistently named miRNAs.

→ This all has been addressed in the manually curated microRNA gene database MirGeneDB.org (15–17)

→ authors should repeat their analyses with MirGeneDB set of microRNAs for *C.elegans*

Deregulation reporting

Figure 1 is poor and should be improved:

Make sure to actually report that this overview figure (1a) is from an earlier publication

Volcano-plot ranges should be standardized

Label microRNAs that are deregulated in volcano plots

Highlight commonly deregulated microRNAs

Check microRNA families

Linking absence of transcription to inter-tissue transport

The authors state absence of detection of transcription equals the actual lack of transcription.

Can the authors rule out that in fact leaky transcription takes place?

Studies of cell lines in human (18) and isolated cell-types in CEL (19) have in depth measured microRNAs. Could the authors use these cell-lines to study the presence of non-transcriptionally active microRNA transcripts?

This has consequences for the EV-work. Please discuss and also check isomir content (but see (20))

Evolutionary interpretations

The authors should attempt to compare the evolutionary age of all microRNAs they find to be age-indicative (which can easily be done in MGDB)

Discuss the indicative evolutionary age for the mechanisms the authors proposed

References

1. Castellano L, Stebbing J. Deep sequencing of small RNAs identifies canonical and non-canonical miRNA and endogenous siRNAs in mammalian somatic tissues. *Nucleic Acids Res* 2013;41:3339–51.
2. Chiang HR, Schoenfeld LW, Ruby JG, Auyeung VC, Spies N, Baek D, Johnston WK, Russ C, Luo S, Babiarz JE, et al. Mammalian microRNAs: experimental evaluation of novel and previously annotated genes. *Genes Dev* 2010;24:992–1009.
3. Jones-Rhoades MW. Conservation and divergence in plant microRNAs. *Plant Mol Biol* 2012;80:3–16.
4. Ludwig N, Becker M, Schumann T, Speer T, Fehlmann T, Keller A, Meese E. Bias in recent miRBase annotations potentially associated with RNA quality issues. *Sci Rep* 2017;7:5162.
5. Langenberger D, Bartschat S, Hertel J, Hoffmann S, Tafer H, Stadler PF. MicroRNA or Not MicroRNA? *Advances in Bioinformatics and Computational Biology*. Springer Berlin Heidelberg; 2011. p. 1–9.

6. Meng Y, Shao C, Wang H, Chen M. Are all the miRBase-registered microRNAs true? A structure- and expression-based re-examination in plants. *RNA Biol* 2012;9:249–53.
7. Tarver JE, Donoghue PC, Peterson KJ. Do miRNAs have a deep evolutionary history? *Bioessays* 2012;34:857–66.
8. Taylor RS, Tarver JE, Hiscock SJ, Donoghue PC. Evolutionary history of plant microRNAs. *Trends Plant Sci* [Internet] 2014; Available from: <http://dx.doi.org/10.1016/j.tplants.2013.11.008>
9. Wang X, Liu XS. Systematic Curation of miRBase Annotation Using Integrated Small RNA High-Throughput Sequencing Data for *C. elegans* and *Drosophila*. *Front Genet* 2011;2:25.
10. Fromm B, Billipp T, Peck LE, Johansen M. A uniform system for the annotation of vertebrate microRNA genes and the evolution of the human microRNAome. *Annual review of [Internet]* [annualreviews.org](http://www.annualreviews.org); 2015; Available from: <http://www.annualreviews.org/doi/abs/10.1146/annurev-genet-120213-092023>
11. Axtell MJ, Meyers BC. Revisiting Criteria for Plant MicroRNA Annotation in the Era of Big Data. *Plant Cell Am Soc Plant Biol*; 2018;30:272–84.
12. Guo Z, Kuang Z, Wang Y, Zhao Y, Tao Y, Cheng C, Yang J, Lu X, Hao C, Wang T, et al. PmiREN: a comprehensive encyclopedia of plant miRNAs. *Nucleic Acids Res* 2020;48:D1114–21.
13. Fromm B, Domanska D, Høy E, Ovchinnikov V, Kang W, Aparicio-Puerta E, Johansen M, Flatmark K, Mathelier A, Hovig E, et al. MirGeneDB 2.0: the metazoan microRNA complement. *Nucleic Acids Res Cold Spring Harbor Laboratory*; 2019;258749.
14. Fromm B, Keller A, Yang X, Friedlander MR, Peterson KJ, Griffiths-Jones S. Quo vadis microRNAs? *Trends Genet* 2020;36:461–3.
15. Fromm B, Billipp T, Peck LE, Johansen M, Tarver JE, King BL, Newcomb JM, Sempere LF, Flatmark K, Hovig E, et al. A Uniform System for the Annotation of Vertebrate microRNA Genes and the Evolution of the Human microRNAome. *Annu Rev Genet* 2015;49:213–42.
16. Fromm B, Domanska D, Høy E, Ovchinnikov V, Kang W, Aparicio-Puerta E, Johansen M, Flatmark K, Mathelier A, Hovig E, et al. MirGeneDB 2.0: the metazoan microRNA complement. *Nucleic Acids Res* 2020;48:D1172.
17. Fromm B, Høy E, Domanska D, Zhong X, Aparicio-Puerta E, Ovchinnikov V, Umu SU, Chabot PJ, Kang W, Aslanzadeh M, et al. MirGeneDB 2.1: toward a complete sampling of all major animal phyla. *Nucleic Acids Res* 2022;50:D204–10.
18. Patil AH, Baran A, Brehm ZP, McCall MN, Halushka MK. A curated human cellular microRNAome based on 196 primary cell types. *Gigascience* [Internet] 2022;11. Available from: <http://dx.doi.org/10.1093/gigascience/giac083>
19. Alberti C, Manzenreither RA, Sowemimo I, Burkard TR, Wang J, Mahofsky K, Ameres SL, Cochella L. Cell-type specific sequencing of microRNAs from complex animal tissues. *Nat Methods* 2018;15:283–9.
20. Gómez-Martín C, Aparicio-Puerta E, van Eijndhoven MAJ, Medina JM, Hackenberg M, Pegtel DM. Reassessment of miRNA variant (isomiRs) composition by small RNA sequencing. *Cell Reports Methods*

[Internet] Elsevier; 2023; Available from: [https://www.cell.com/cell-reports-methods/pdf/S2667-2375\(23\)00103-0.pdf](https://www.cell.com/cell-reports-methods/pdf/S2667-2375(23)00103-0.pdf)

Reviewer #2 (Remarks to the Author):

This study profiled age-dependent changes in transcriptome, miRNAome expression across five somatic tissues in the nematode *C. elegans*. The author also profiled EVs composition and identified several miRNAs present in EVs not transcribed in other tissues. The author tested one of them (miR-1) and its relationship with the intestine gene DAF-16. The results of this analysis and the main experiments shown let the author conclude that many miRNAs are indeed transported from one tissue to another during the worm's lifespan.

This very interesting topic is not novel, but this study provides an updated view of the crosstalk between EV miRNAs and somatic tissue miRNAs. Overall, the manuscript is good, but it lacks wet-bench validation throughout it. In addition, some of the figures and tables need to be revised to convey their message better.

Major issues:

- Similar transcriptomes and miRNA isolation techniques have been used in the past and must be used as a comparison. Compare miRNA data with Schorr et al., 2023 and Serizay et al, 2020 and Alberti et al. 2018.
- Where is the raw data used to generate Figure 3? Did the author prepare the 65 promoter::GFP constructs, prepare transgenic strains and study their fluorescence pattern? Did they use previously published data? This is unclear and needs to be addressed.
- In the discussion section, Figure 3 shows 19 miRNAs with a PmiR:GFP signal but no miRNA-seq signal. The authors should compare the expression of these miRNAs in other published datasets.
- In Extended Data Figure 3d, some of the PITT-miRs not detected in EVs have high UMIs. Why are these highly expressed PITT-miRs not being detected in EVs?

- Technical issue: Are there similarities among the cells used for each tissue? The principal component analysis suggests that. The mechanical separation used by the authors may have released the same cells in each tissue, which may not be a good representative of each tissue's complexity itself. This point at least needs to be mentioned in the discussion section.
- Figure 4 is misleading because the author only sequenced five tissues and did not have a global transcriptome/miRNAome for all tissues. It could be that a given miRNA is transported from a different tissue not sequenced by the authors.
- The model proposed by the author is fascinating but is not adequately discussed in the discussion section. Do worm somatic cells have a check in place for each tissue to absorb only one kind of EV containing specific miRNAs? This is highly unlikely, and its function needs to be adequately discussed and speculated in the discussion section.
- It is unclear how EVs were isolated. In the method section, the authors mentioned QIAzol, which is used to isolate total RNAs and differential centrifuges. Could there be contamination from total RNAs, which is not representative of EV RNAs? This critical step needs to be explained in the method section.
- The author did not address from which tissue EVs were released. This initial part of the manuscript is purely bioinformatics, with no wet bench validation.
- The daf-16/miR-1 section is very nice, but the author did not show the critical point that miR-1 found in the intestine is of muscle tissue origin. Although transcribed in the muscle, several studies found this miRNA also in other tissues. Perhaps the author could label muscle miRNAs and see if they detect in the intestine (mime-seq?).
- Is there an enrichment of age-related targets for Age-DEMIRs? Please repeat this analysis with non-Age-DEMIRs and include it in Figure 6c.
- Figure 7a does not offer much insight into how inter-tissue miRNA signaling regulates aging. Please replace it with a figure or table showing which mRNAs are targeted by PITT-miRs in day 1 worms and in day 8 worms (and if there is any difference between these two groups).

Minor issues:

- The legend for Figures 6a and 6b must be in separate paragraphs.
- Figure 6b does not offer much insight into how miRNAs regulate aging. Please replace it with a figure or table showing which mRNAs are targeted by miRNAs in day 1 worms and day 8 worms (and if there is any difference between these two groups).
- In Extended Data Figure 3d, many PITT-miRs not detected in EVs appear to have a mean UMI of 0. However, the shown threshold for expression (and thus inclusion in your analyses) is a UMI sum of 10. Is this a byproduct of how this figure was made?
- Extended Data Table 2 is unclear and does not explain how tissue-specific miRNAs change during aging. Please revise this table by including the specific EV miRNAs that changed with age.

We are grateful to the reviewers for their time and efforts on our manuscript. We appreciate their insightful and constructive comments. Below is our point-to-point response.

Our replies in the rebuttal letter and corresponding modifications in the revised main text are marked in blue.

Reviewer #1 (Remarks to the Author):

REVIEW NCOMMS-23-20619

The manuscript: "Tissue-specific profiling of age-dependent miRNAomic changes in *Caenorhabditis elegans*" is conceptually very interesting. For the main claims of this manuscript - inter-tissue trafficking as measured by the lack of transcription but the presence of microRNAs - however, I see substantial shortcomings. Many aspects of the study are too briefly explained and conclusions too bold in the light of the presented analyses /results.

Use of miRBase:

For years, a major concern in microRNA research has been the quality of the online repository miRBase (1–14) with estimates of 2/3 false-positive entries. Thus, the database contains many false-positives and not only microRNAs. These are for instance numerous tRNA, rRNA or other fragments and transcriptional noise but also incorrectly annotated bona fide miRNAs that strongly influence interpretations of data. In addition to the false positives, miRBase annotations are often imprecise and have varying precursor annotation forms (with or without flanking regions of varying lengths) and very often not both arms are annotated, 3' ends are incorrect, and in a few cases even 5' are not correctly annotated which substantially affects target predictions, and more so, in this case here when needing accurately defined Drosha and Dicer cutting sites. Further it is using an outdated nomenclature which is inconsistent and leads to problems of not-summarizing inconsistently named miRNAs.

This all has been addressed in the manually curated microRNA gene database MirGeneDB.org (15–17)

→ authors should repeat their analyses with MirGeneDB set of microRNAs for *C. elegans*

-- As suggested, all our analyses have been re-performed by the *C. elegans* miRNAs curated in MirGeneDB 2.1. In our previous analysis of inter-tissue miRNA signalling, only miRNAs with high confidence were considered. These miRNAs are all in MirGeneDB, except for one miRNA. So, our conclusions of inter-tissue miRNA signalling are barely affected.

Deregulation reporting

Figure 1 is poor and should be improved:

Make sure to actually report that this overview figure (1a) is from an earlier publication

-- Fig. 1a in the initial submission was from our earlier publication with slight modification (Wang et al., 2022). In the revised manuscript, we have replaced the previous diagram with a newly created one.

Volcano-plot ranges should be standardized

-- The volcano-plot ranges have been standardized in the revised Fig 1b, with \log_2FC from -8 to 8 and $\log_{10}P$ from 0 to 8.

Label microRNAs that are deregulated in volcano plots

-- Representative Age-DEMIRs are labelled in volcano plots as suggested.

Highlight commonly deregulated microRNAs

-- Thank you for your suggestion! The Age-DEMIRs shared in multiple tissues are listed in the revised Extended Data Table 1 (sheet: 'shared across tissues').

Check microRNA families

-- The information of microRNA families has been included in the revised Extended Data Table 1.

Linking absence of transcription to inter-tissue transport

The authors state absence of detection of transcription equals the actual lack of transcription.

Can the authors rule out that in fact leaky transcription takes place?

-- We analysed transcription using PmiR::GFP reporters. Due to the sensitivity of these reporters, we can hardly rule out that in fact leaky transcription could take place. Therefore, we have not stated that absence of detection of transcription equals the actual lack of transcription. We further clarified this issue in the revised manuscript.

Our datasets suggest that a group of miRNAs are highly likely to be transported across worm tissues by the discrepancies in their PmiR::GFP and miRNA-Seq signals. We term them as PITT-miRNA (Potentially Inter-Tissue Transported miRNA).

Studies of cell lines in human (18) and isolated cell-types in CEL (19) have in depth measured microRNAs. Could the authors use these cell-lines to study the presence of non-transcriptionally active microRNA transcripts?

This has consequences for the EV-work.

-- There are extensive studies on secreted miRNAs in EV, using cultured cell lines (including human cell lines). Based on these studies, it is now well-accepted that EV-encapsulated miRNAs are inter-cellular messengers (Maas et al., 2017). However, little is known about the scale of inter-tissue miRNA signalling, which are potentially mediated by EV, in a living organism. To address this issue, this manuscript aims at providing a global picture of inter-tissue miRNA signalling, not only at an organismal (*C. elegans*) level but also from a miRNAomic perspective.

To achieve this goal, we did use freshly isolated worm cells, although not worm cell lines, to profile tissue-specific miRNAomes (Figure 1). Combined with a systematic analysis of PmiR::GFP signal in worm tissues, we then mapped the inter-tissue miRNA trafficking network based on PITT-miRs (Figure 3 and 4). Our previous report on *mir-83* (Zhou et al., 2019) and the *mir-1*-related studies in this manuscript (Fig. 8 and Extended Data Fig. 7) provide further evidence to support this network.

Please discuss and also check isomir content (but see (20))

-- The IsoSeek method, as employed in reference 20, utilizes a randomized-end adapter strategy to mitigate ligation bias, thereby enabling the detection of isomiRs. Several analogous methods have been developed since 2015, such as 4N-seq and AQ-seq. While this approach is powerful, none of these techniques are capable of profiling small RNAs using < 1 ng of total RNA.

In our study, our small RNA library construction relies on a minute sample of

approximately 20 cells, equivalent to 100-200 pg of total RNA. Consequently, we were unable to employ the randomized-end adapter-based method to assess and quantify isomiR content. We acknowledge that the inclusion of isomiR-related information in our study could provide valuable insights into the functional roles of miRNAs in aging. We remain optimistic that ongoing advancements in small RNA sequencing methodologies will eventually overcome this limitation, broadening the scope of such research. In the revised manuscript, we discuss the isomiR issue as suggested. Please see page 6 and 23 for details.

Evolutionary interpretations

The authors should attempt to compare the evolutionary age of all microRNAs they find to be age-indicative (which can easily be done in MGDB).

-- Thank you for your suggestion! In the revised manuscript, we labelled the evolutionary age of all examined microRNAs in Extended Data Table 1 by the information in MirGeneDB.

Discuss the indicative evolutionary age for the mechanisms the authors proposed.

-- The evolutionary age of Age-DEMIRs does not show a clear difference from that of all detected miRNAs. Of interest, PITT-miRs are modestly older in evolution than all detected miRNAs, suggesting that inter-cellular miRNA trafficking could be an ancient way of cell-to-cell communication. Corresponding discussion has been included in the revised manuscript. Please see page 19 for details.

References

1. Castellano L, Stebbing J. Deep sequencing of small RNAs identifies canonical and non-canonical miRNA and endogenous siRNAs in mammalian somatic tissues. *Nucleic Acids Res* 2013;41:3339–51.
2. Chiang HR, Schoenfeld LW, Ruby JG, Auyeung VC, Spies N, Baek D, Johnston WK, Russ C, Luo S, Babiarz JE, et al. Mammalian microRNAs: experimental evaluation of novel and previously annotated genes. *Genes Dev* 2010;24:992–1009.
3. Jones-Rhoades MW. Conservation and divergence in plant microRNAs. *Plant Mol Biol* 2012;80:3–16.
4. Ludwig N, Becker M, Schumann T, Speer T, Fehlmann T, Keller A, Meese E. Bias in recent miRBase annotations potentially associated with RNA quality issues. *Sci Rep*

2017;7:5162.

5. Langenberger D, Bartschat S, Hertel J, Hoffmann S, Tafer H, Stadler PF. MicroRNA or Not MicroRNA? *Advances in Bioinformatics and Computational Biology*. Springer Berlin Heidelberg; 2011. p. 1–9.
6. Meng Y, Shao C, Wang H, Chen M. Are all the miRBase-registered microRNAs true? A structure- and expression-based re-examination in plants. *RNA Biol* 2012;9:249–53.
7. Tarver JE, Donoghue PC, Peterson KJ. Do miRNAs have a deep evolutionary history? *Bioessays* 2012;34:857–66.
8. Taylor RS, Tarver JE, Hiscock SJ, Donoghue PC. Evolutionary history of plant microRNAs. *Trends Plant Sci* [Internet] 2014; Available from: <http://dx.doi.org/10.1016/j.tplants.2013.11.008>
9. Wang X, Liu XS. Systematic Curation of miRBase Annotation Using Integrated Small RNA High-Throughput Sequencing Data for *C. elegans* and *Drosophila*. *Front Genet* 2011;2:25.
10. Fromm B, Billipp T, Peck LE, Johansen M. A uniform system for the annotation of vertebrate microRNA genes and the evolution of the human microRNAome. *Annual review of* [Internet] annualreviews.org; 2015; Available from: <http://www.annualreviews.org/doi/abs/10.1146/annurev-genet-120213-092023>
11. Axtell MJ, Meyers BC. Revisiting Criteria for Plant MicroRNA Annotation in the Era of Big Data. *Plant Cell Am Soc Plant Biol*; 2018;30:272–84.
12. Guo Z, Kuang Z, Wang Y, Zhao Y, Tao Y, Cheng C, Yang J, Lu X, Hao C, Wang T, et al. PmiREN: a comprehensive encyclopedia of plant miRNAs. *Nucleic Acids Res* 2020;48:D1114–21.
13. Fromm B, Domanska D, Høye E, Ovchinnikov V, Kang W, Aparicio-Puerta E, Johansen M, Flatmark K, Mathelier A, Hovig E, et al. MirGeneDB 2.0: the metazoan microRNA complement. *Nucleic Acids Res Cold Spring Harbor Laboratory*; 2019;258749.
14. Fromm B, Keller A, Yang X, Friedlander MR, Peterson KJ, Griffiths-Jones S. Quo vadis microRNAs? *Trends Genet* 2020;36:461–3.
15. Fromm B, Billipp T, Peck LE, Johansen M, Tarver JE, King BL, Newcomb JM, Sempere LF, Flatmark K, Hovig E, et al. A Uniform System for the Annotation of Vertebrate microRNA Genes and the Evolution of the Human microRNAome. *Annu Rev Genet* 2015;49:213–42.
16. Fromm B, Domanska D, Høye E, Ovchinnikov V, Kang W, Aparicio-Puerta E, Johansen M, Flatmark K, Mathelier A, Hovig E, et al. MirGeneDB 2.0: the metazoan microRNA complement. *Nucleic Acids Res* 2020;48:D1172.
17. Fromm B, Høye E, Domanska D, Zhong X, Aparicio-Puerta E, Ovchinnikov V, Umu SU, Chabot PJ, Kang W, Aslanzadeh M, et al. MirGeneDB 2.1: toward a complete sampling of all major animal phyla. *Nucleic Acids Res* 2022;50:D204–10.

18. Patil AH, Baran A, Brehm ZP, McCall MN, Halushka MK. A curated human cellular microRNAome based on 196 primary cell types. *Gigascience* [Internet] 2022;11. Available from: <http://dx.doi.org/10.1093/gigascience/giac083>
19. Alberti C, Manzenreither RA, Sowemimo I, Burkard TR, Wang J, Mahofsky K, Ameres SL, Cochella L. Cell-type specific sequencing of microRNAs from complex animal tissues. *Nat Methods* 2018;15:283–9.
20. Gómez-Martín C, Aparicio-Puerta E, van Eijndhoven MAJ, Medina JM, Hackenberg M, Pegtel DM. Reassessment of miRNA variant (isomiRs) composition by small RNA sequencing. *Cell Reports Methods* [Internet] Elsevier; 2023; Available from: [https://www.cell.com/cell-reports-methods/pdf/S2667-2375\(23\)00103-0.pdf](https://www.cell.com/cell-reports-methods/pdf/S2667-2375(23)00103-0.pdf)

Reviewer #2 (Remarks to the Author):

This study profiled age-dependent changes in transcriptome, miRNAome expression across five somatic tissues in the nematode *C. elegans*. The author also profiled EVs composition and identified several miRNAs present in EVs not transcribed in other tissues. The author tested one of them (miR-1) and its relationship with the intestine gene DAF-16. The results of this analysis and the main experiments shown let the author conclude that many miRNAs are indeed transported from one tissue to another during the worm's lifespan.

This very interesting topic is not novel, but this study provides an updated view of the crosstalk between EV miRNAs and somatic tissue miRNAs. Overall, the manuscript is good, but it lacks wet-bench validation throughout it. In addition, some of the figures and tables need to be revised to convey their message better.

Major issues:

- Similar transcriptomes and miRNA isolation techniques have been used in the past and must be used as a comparison. Compare miRNA data with Schorr et al., 2023 and Serizay et al, 2020 and Alberti et al. 2018.

-- Thank you for your suggestion! We have looked up the miRNA data in the three reports you mentioned. Whereas the reports by 'Alberti et al., 2018' and 'Schorr et al., 2023' analysed miRNA in worm tissues (Alberti et al., 2018; Schorr et al., 2023), the study by 'Serizay et al., 2020' focused on protein-coding genes (Serizay et al., 2020). In addition, as discussed in our initial submission, Brosnan et al. profiled tissue-specific miRNAome by immunoprecipitating Argonaut (Brosnan et al., 2021). Therefore, we compared our results with the data from these three reports in the revised manuscript (Alberti et al., 2018; Brosnan et al., 2021; Schorr et al., 2023), as suggested. Please see page 6 and 17-18, Extended Data Fig 1, and Extended Data Table 2 for details.

In short, our datasets overlap with previous ones but with more miRNAs detected. This could be due to the following two reasons:

- a. We preformed miRNA-Seq directly on worm cells, without additional steps of immunoprecipitation, RNA modification, or nuclear isolation. Our method thus should have a higher detection sensitivity.
- b. Our analysis is on worms at day 1 or 8 adulthood, whereas previous studies on

larvae or worms of mixed stages. As many worm miRNAs exhibit specific temporal expression pattern, the different timing of analysed samples could make the results vary from each other.

- Where is the raw data used to generate Figure 3? Did the author prepare the 65 promoter::GFP constructs, prepare transgenic strains and study their fluorescence pattern? Did they use previously published data? This is unclear and needs to be addressed.

-- The 65 transgenic strains were prepared in the previous studies (de Lencastre et al., 2010; Martinez et al., 2008; Xu et al., 2019) and obtained from Caenorhabditis Genetics Center (CGC). Because their published expression patterns were not checked in aged worms, we, by ourselves, examined their expression in the five worm tissues at both day 1 and day 8 of adulthood. Corresponding reports were cited in our initially submitted manuscript. In the revised manuscript, we have further clarified this issue. Please see page 8.

The raw data of miRNA-Seq data in Figure 3 have been uploaded to SRA database, as shown in Data Availability. The raw data of fluorescent images are hard to upload to a public database due to their size. We are glad to share these images upon request.

- In the discussion section, Figure 3 shows 19 miRNAs with a PmiR::GFP signal but no miRNA-seq signal. The authors should compare the expression of these miRNAs in other published datasets.

-- Thank you for your suggestion! Because we have reanalysed our data by the curation in MirGeneDB in the revised manuscript, as suggested by Reviewer 1, there are now 11 miRNA genes with a *PmiR::GFP* signal but no miRNA-Seq signal in at least one of the examined tissues. These 11 miRNAs are excluded in our PITT-miR analysis.

We have compared their expression in other three published datasets (Alberti et al., 2018; Brosnan et al., 2021; Schorr et al., 2023), as suggested. Please see page 8-9 and Extended Data Table 2 (Sheet 2) for details. In short, 4 of the 11 miRNAs are not detected in any of the three published datasets, whereas the detection of the rest miRNAs in these datasets varies. This could be due to the differences in sample preparation and ages of analysed worms.

- In Extended Data Figure 3d, some of the PITT-miRs not detected in EVs have high UMIs. Why are these highly expressed PITT-miRs not being detected in EVs?

-- Good question! We think this shows that there could be other ways for miRNAs to be transported across tissues. According to previous reports, circulating RNAs are not necessarily EV-associated (Cui et al., 2019; Kumar and Reddy, 2016). Corresponding discussion has been included in the revised discussion section. Please see page 19 for details.

- Technical issue: Are there similarities among the cells used for each tissue? The principal component analysis suggests that.

-- We appreciate your notice of this similarity. Although these cells are from different tissues, they should share some basic biological processes. In fact, our previous report on tissue-specific mRNA transcriptomes clearly shows this fact (Wang et al., 2022). The gene set enrichment analysis (WormCat) in this study also indicate that miRNAs in distinct tissues are regulating some common activities. In the revised manuscript, we have further clarified this issue in the revised manuscript. Please see page 7 for details.

The mechanical separation used by the authors may have released the same cells in each tissue, which may not be a good representative of each tissue's complexity itself. This point at least needs to be mentioned in the discussion section.

-- It is a feature of *C. elegans* tissues, except for neurons and germline, that they consist of highly homogeneous cells. Therefore, the dozens of hand-picked cells from intestine, body wall muscle, hypodermis, and coelomocytes can well represent these tissues. For neurons, we collected thousands of neurons following the protocol established by Murphy lab (Kaletsky et al., 2015). Given the number of analysed neurons in this study, our neuronal dataset should cover different types of worm neurons and be representative of this tissue.

Yet, we also agree with the reviewer that the highly homogeneous cells in a worm tissue could still have slight variations. Corresponding discussion have been added in the revised manuscript. Please see page 17.

- Figure 4 is misleading because the author only sequenced five tissues and did not have a global transcriptome/miRNAome for all tissues. It could be that a given miRNA is

transported from a different tissue not sequenced by the authors.

-- We appreciate your notice of this possibility that a given miRNA could be transported from a non-sequenced tissue into the five analysed ones. In our initial submission, we showed this situation in Figure 4 with brown arrows, although without highlighting. In the revised manuscript, we further clarified this issue in the Result and Discussion sections (page 10 and 20), Figure 4 and its legend.

- The model proposed by the author is fascinating but is not adequately discussed in the discussion section. Do worm somatic cells have a check in place for each tissue to absorb only one kind of EV containing specific miRNAs? This is highly unlikely, and its function needs to be adequately discussed and speculated in the discussion section.

-- Many thanks for your appreciation of our model! We have included the suggested issues in the revised Discussion section. Please see page 20 for details.

- It is unclear how EVs were isolated. In the method section, the authors mentioned QIAzol, which is used to isolate total RNAs and differential centrifuges. Could there be contamination from total RNAs, which is not representative of EV RNAs? This critical step needs to be explained in the method section.

-- Sorry for the confusion! We isolated EVs secreted by worms following the established protocol reported by both us and Kaeberlein lab (Russell et al., 2020; Zhou et al., 2019). In brief, worms were cultured in M9 buffer for 6 h. While the sedimented worms were collected as worm samples, the supernatant was subjected to ultracentrifugation to isolate the EVs secreted from worms. QIAzol was used in the subsequent RNA preparations for both worm and EV samples.

Therefore, it is highly unlikely that there is any contamination from worm total RNAs in EV RNAs. We have further clarified this critical issue in the Method section as suggested. Please see page 26 for details.

- The author did not address from which tissue EVs were released.

-- As mentioned above, we collected EVs secreted from worms. Therefore, we are unclear which tissues EVs are from. That is why we compared the miRNAome in EV with

that in the whole worm (Figure 5), but not with those in specific worm tissues.

We agree with the reviewer that it is interesting to map the precise source tissues of these EVs. It is likely that different tissues could secrete distinct EVs. Although it is now hard to address this issue due to technical hinderances, we believe it will be a fascinating and critical topic to pursue in the future. Corresponding discussion has been included in the revised manuscript. Please see page 21 for details.

This initial part of the manuscript is purely bioinformatics, with no wet bench validation.

- The *daf-16*/miR-1 section is very nice, but the author did not show the critical point that miR-1 found in the intestine is of muscle tissue origin. Although transcribed in the muscle, several studies found this miRNA also in other tissues. Perhaps the author could label muscle miRNAs and see if they detect in the intestine (mime-seq?).

-- Using a transgene expressing miR-1 specifically in body wall muscle and a mutant worm strain knocked out of the endogenous *mir-1* gene, we detected muscle-derived miR-1 in intestine by RT-qPCR (Extended Data Fig. 7b). Although we did not analyse other tissues, we believe that miR-1 in other non-muscle tissues, as the reviewer mentioned, is likely to be of muscle origin, by our inter-tissue miRNA analysis (Figure 3 and 4).

- Is there an enrichment of age-related targets for Age-DEMIRs? Please repeat this analysis with non-Age-DEMIRs and include it in Figure 6c.

-- By the genes reported in GenAge, we do not observe an enrichment of age-related targets for Age-DEMIRs. The ratio of GenAge-curated genes in Age-DEMIR targets is similar to that in non-Age-DEMIR targets across the five examined tissues (Table 1 for reviewer).

Tissue	Non-Age-DEMIRs			Age-DEMIRs		
	# Targets	# GenAge Targets	% GenAge Targets	# Targets	# GenAge Targets	% GenAge Targets
Neuron	1618	137	8.47%	1165	97	8.33%
Intestine	1579	143	9.06%	610	54	8.85%
BWM	1485	134	9.02%	582	54	9.28%
Hypodermis	1793	156	8.70%	1077	96	8.91%
Coelomocyte	1521	131	8.61%	504	54	10.71%

Table 1 for reviewer. A comparison of the age-related targets of Age-DEMIRs and non-Age-DEMIRs. The targets reported in GenAge are considered as age-related.

Yet, no enrichment of GenAge targets for Age-DEMIRs does not necessarily suggest that Age-DEMIRs do not regulate ageing. First, GenAge does not cover all age-related genes. It is already a strong implication of AgeDEMIR-dependent regulation in ageing that 8-9% of their targets are reported in GenAge. Moreover, Age-DEMIRs are the miRNAs changed during ageing, whereas non-Age-DEMIRs are not. Although non-Age-DEMIRs could also target many age-related genes, they are unlikely to regulate ageing as actively as Age-DEMIRs.

This study aims at the age-dependent miRNAomic changes in worm tissues. Therefore, the analysis of age-related genes in non-Age-DEMIR targets is out of the scope of this manuscript, from our point of view. To help the readers focus on the major topic of this study, we would like to present this analysis in the rebuttal letter, which will also be published online along with the manuscript, instead of including it in Figure 6c.

- Figure 7a does not offer much insight into how inter-tissue miRNA signaling regulates aging. Please replace it with a figure or table showing which mRNAs are targeted by PITT-miRs in day 1 worms and in day 8 worms (and if there is any difference between these two groups).

-- Thank you for your suggestion! Figure 7a presents the number of potential PITT-miRs targets, aiming to show a general scale of the regulation by PITT-miRs. The PITT-miRs targets are listed in Extended Data Table 4 (sheets 'Day 1 PITT-miRs' and 'Day 8 PITT-miRs').

In the revised manuscript, we have further compared the two groups of PITT-miRs targets as suggested. Please see the revised Extended Data Fig. 5 for details.

Minor issues:

- The legend for Figures 6a and 6b must be in separate paragraphs.

-- The legend has been modified according to your comment.

- Figure 6b does not offer much insight into how miRNAs regulate aging. Please replace it with a figure or table showing which mRNAs are targeted by miRNAs in day 1 worms and day 8 worms (and if there is any difference between these two groups).

-- Thank you for your suggestion! Figure 6b shows the number of potential miRNA targets, aiming to give readers a general idea of the scale of miRNA-dependent regulation. All predicted miRNA targets are listed in Extended Data Table 4 (sheets 'Day 1 All-miRs' and 'Day 8 All-miRs').

In the revised manuscript, we have further compared the two groups of miRs targets as suggested. Please see page 13 and the revised Extended Data Fig. 5 for details.

- In Extended Data Figure 3d, many PITT-miRs not detected in EVs appear to have a mean UMI of 0. However, the shown threshold for expression (and thus inclusion in your analyses) is a UMI sum of 10. Is this a byproduct of how this figure was made?

-- In Extended Data Figure 3d (Extended Data Figure 4d in the revised manuscript), we compared the expression of PITT-miRs detected and not detected in EVs in our miRNA-Seq of whole worms. As you noticed, many of those not detected in EVs have a mean UMI of 0. These miRNAs are excluded in our analysis in Figure 5 and Extended Data Table 3 because the threshold of these analyses is a UMI sum of 10.

However, a UMI sum below 10 indicates their low expression. So, although excluded in other analysis, we presented these miRNAs in Extended Data Figure 3d to show the huge difference of expression levels between PITT-miRs detected and not detected in EVs. We have further clarified this issue in the corresponding figure legend.

- Extended Data Table 2 is unclear and does not explain how tissue-specific miRNAs change during aging. Please revise this table by including the specific EV miRNAs that changed with age.

-- The EV and worm miRNAs that change with age are highlighted in the revised Extended Data Table 3 (previously Extended Data Table 2), as suggested.

References

- Alberti, C., Manzenreither, R.A., Sowemimo, I., Burkard, T.R., Wang, J., Mahofsky, K., Ameres, S.L., and Cochella, L. (2018). Cell-type specific sequencing of microRNAs from complex animal tissues. *Nature methods* *15*, 283-289.
- Brosnan, C.A., Palmer, A.J., and Zuryn, S. (2021). Cell-type-specific profiling of loaded miRNAs from *Caenorhabditis elegans* reveals spatial and temporal flexibility in Argonaute loading. *Nature communications* *12*, 2194.
- Cui, M., Wang, H., Yao, X., Zhang, D., Xie, Y., Cui, R., and Zhang, X. (2019). Circulating MicroRNAs in Cancer: Potential and Challenge. *Frontiers in genetics* *10*, 626.
- de Lencastre, A., Pincus, Z., Zhou, K., Kato, M., Lee, S.S., and Slack, F.J. (2010). MicroRNAs both promote and antagonize longevity in *C. elegans*. *Current biology : CB* *20*, 2159-2168.
- Kaletsky, R., Lakhina, V., Arey, R., Williams, A., Landis, J., Ashraf, J., and Murphy, C.T. (2015). The *C. elegans* adult neuronal IIS/FOXO transcriptome reveals adult phenotype regulators. *Nature*.
- Kumar, S., and Reddy, P.H. (2016). Are circulating microRNAs peripheral biomarkers for Alzheimer's disease? *Biochim Biophys Acta* *1862*, 1617-1627.
- Maas, S.L.N., Breakefield, X.O., and Weaver, A.M. (2017). Extracellular Vesicles: Unique Intercellular Delivery Vehicles. *Trends in cell biology* *27*, 172-188.
- Martinez, N.J., Ow, M.C., Reece-Hoyes, J.S., Barrasa, M.I., Ambros, V.R., and Walhout, A.J. (2008). Genome-scale spatiotemporal analysis of *Caenorhabditis elegans* microRNA promoter activity. *Genome research* *18*, 2005-2015.
- Russell, J.C., Kim, T.K., Noori, A., Merrihew, G.E., Robbins, J.E., Golubeva, A., Wang, K., MacCoss, M.J., and Kaeberlein, M. (2020). Composition of *Caenorhabditis elegans* extracellular vesicles suggests roles in metabolism, immunity, and aging. *GeroScience* *42*, 1133-1145.
- Schorr, A.L., Mejia, A.F., Miranda, M.Y., and Mangone, M. (2023). An updated *C. elegans* nuclear body muscle transcriptome for studies in muscle formation and function. *Skelet Muscle* *13*, 4.
- Serizay, J., Dong, Y., Janes, J., Chesney, M., Cerrato, C., and Ahringer, J. (2020). Distinctive regulatory architectures of germline-active and somatic genes in *C. elegans*. *Genome research* *30*, 1752-1765.
- Wang, X., Jiang, Q., Song, Y., He, Z., Zhang, H., Song, M., Zhang, X., Dai, Y., Karalay, O., Dieterich, C., *et al.* (2022). Ageing induces tissue-specific transcriptomic changes in *Caenorhabditis elegans*. *The EMBO journal* *41*, e109633.
- Xu, Y., He, Z., Song, M., Zhou, Y., and Shen, Y. (2019). A microRNA switch controls dietary restriction-induced longevity through Wnt signaling. *EMBO reports*.
- Zhou, Y., Wang, X., Song, M., He, Z., Cui, G., Peng, G., Dieterich, C., Antebi, A., Jing, N., and Shen, Y. (2019). A secreted microRNA disrupts autophagy in distinct tissues of *Caenorhabditis elegans* upon ageing. *Nature communications* *10*, 4827.

REVIEWER COMMENTS

Reviewer #1 (Remarks to the Author):

The authors addressed most of my comments and, I believe also of reviewer 2.

In addition to my previous comments, I would like to ask the authors for

- a presentation of the extraction and sequencing results (Please also provide the RNA extraction method!): Please provide a supplementary table with sequenced reads, % microRNA reads and potential contamination - easiest would be to use miRTrace (Kang et al) on the fastqs. Apropos fastqs - I was not able to access the raw data as it was not yet released - please make the data accessible.

- Summarize how many mRNAs you detect in how many cells of the smartseq data.

- Please explain the statement "All other data are available from the corresponding author upon reasonable request."

Reviewer #2 (Remarks to the Author):

The authors have satisfactorily addressed all of my concerns.

We sincerely thank the reviewers for their evaluation of our revised manuscript. We are pleased to note their positive remarks regarding our revisions. We have addressed their remaining concerns in a point-to-point response below. Our responses and corresponding changes in the manuscript are highlighted in blue for clarity.

REVIEWER COMMENTS

Reviewer #1 (Remarks to the Author):

The authors addressed most of my comments and, I believe also of reviewer 2.

-- We are glad to learn your appreciation of our revised manuscript.

In addition to my previous comments, I would like to ask the authors for

- a presentation of the extraction and sequencing results (Please also provide the RNA extraction method!): Please provide a supplementary table with sequenced reads, % microRNA reads and potential contamination - easiest would be to use miRTrace (Kang et al) on the fastqs. Apropos fastqs

-- As suggested, the revised manuscript has included a new supplementary table (Supplementary Table 9). miRTrace was used following your suggestion¹. We hope this table can provide our readers with a comprehensive view of our miRNA-Seq of worm cells. Thank you for your suggestion!

Our approach involves the lysis of 5-10 worm cells (cells from tissues other than neurons) to construct small RNA sequencing libraries. Consequently, the observed percentage of miRNA reads of these samples may appear lower compared to standard bulk sequencing methodologies. However, our miRNA-sequencing of isolated worm cells is of at least similar quality to another recent report of single-cell miRNA-Seq² (Fig. 1 for reviewer).

Fig. 1 for reviewer

Fig. 1 for reviewer. The total reads and miRNA reads in this study and a previous report by Faridani et al. Neu: neuron, Int: intestine, BWM: body wall muscle, Hyp: hypodermis, Coel: coelomocytes.

The RNA extraction method has been detailed in the revised ‘Small RNA sequencing of isolated cells from worm tissues’ in ‘Methods’.

- I was not able to access the raw data as it was not yet released - please make the data accessible.

-- We are sorry for the trouble! These raw data were previously provided with reviewer links, which were in two lines. By submitting our newly revised manuscript, we have made these data accessible to everyone. Please use the following links:

- Tissue miRNA-Seq: <https://www.ncbi.nlm.nih.gov/sra/?term=PRJNA854735>
- Worm and EV miRNA-Seq: <https://www.ncbi.nlm.nih.gov/sra/?term=PRJNA868152>

Please also see ‘Data Availability’ in the revised manuscript. If you have difficulty accessing these datasets, please get in touch with us through Nature Communications or directly at yidong.shen@sibcb.ac.cn at your earliest convenience.

- Summarize how many mRNAs you detect in how many cells of the smartseq data.

-- This information has also been included in the new Supplementary Table 9. The mRNA-Seq data of isolated worm cells are from our published report³.

- Please explain the statement "All other data are available from the corresponding author upon

reasonable request."

-- All miRNA-Seq datasets are deposited in the SRA database with the links listed above.

The tissue-specific mRNA-Seq datasets used in this manuscript are from our published report³.

The corresponding description of the source of mRNA-Seq datasets could be confusing in the 'Result' section. The text has been revised to clarify this issue. Please see page 12. They have been available at the SRA database since the publication of the corresponding report³:

- Whole worm RNA-Seq: <https://www.ncbi.nlm.nih.gov/sra/?term=PRJNA759704>.
- Tissues Smart-Seq2: <https://www.ncbi.nlm.nih.gov/sra/?term=PRJNA759083>.

The worm strains and RT-qPCR primer used in this study are listed in Supplementary Table 8.

All plasmids constructed in this study have been submitted to BRICS (<http://www.brics.ac.cn/plasmid/template/article/about.html>).

We are glad to provide other data not in these public repositories and detailed information of materials and methods in this manuscript upon reasonable request. For example, we sent '.sam' and '.bam' output files of our datasets in the report of 'Wang et al., EMBO 2022'³, to a requester through google drive this September.

We are also glad to share the worm strains and other non-commercial materials used in this study.

Reviewer #2 (Remarks to the Author):

The authors have satisfactorily addressed all of my concerns.

-- Thank you again for your time and effort on our manuscript!

References

1. Kang, W., Eldfjell, Y., Fromm, B., Estivill, X., Biryukova, I., and Friedlander, M.R. (2018). miRTrace reveals the organismal origins of microRNA sequencing data. *Genome biology* *19*, 213. [10.1186/s13059-018-1588-9](https://doi.org/10.1186/s13059-018-1588-9).
2. Faridani, O.R., Abdullayev, I., Hagemann-Jensen, M., Schell, J.P., Lanner, F., and Sandberg, R. (2016). Single-cell sequencing of the small-RNA transcriptome. *Nature biotechnology* *34*, 1264-1266. [10.1038/nbt.3701](https://doi.org/10.1038/nbt.3701).
3. Wang, X., Jiang, Q., Song, Y., He, Z., Zhang, H., Song, M., Zhang, X., Dai, Y., Karalay, O., Dieterich, C., et al. (2022). Ageing induces tissue-specific transcriptomic changes in *Caenorhabditis elegans*. *The EMBO journal* *41*, e109633. [10.15252/embj.2021109633](https://doi.org/10.15252/embj.2021109633).